# Functional annotation of the Hippo pathway somatic mutations in human cancers

Han Han [1,2,3,14] ✉, Zhen Huang[4,14], Congsheng Xu[5,14], Gayoung Seo[3], Jeongmin An [3], Bing Yang[3], Yuhan Liu[3], Tian Lan[3], Jiachen Yan[3], Shanshan Ren[5], Yue Xu[1,2], Di Xiao[1,2], Jonathan K. Yan[3], Claire Ahn[3], Dmitry A. Fishman [6], Zhipeng Meng [7], Kun-Liang Guan [8], Ruxi Qi[9] ✉, Ray Luo [10,11,12,13] ✉ & Wenqi Wang [3] ✉

The Hippo pathway is commonly altered in cancer initiation and progression; however, exactly how this pathway becomes dysregulated to promote human cancer development remains unclear. Here we analyze the Hippo somatic mutations in the human cancer genome and functionally annotate their roles in targeting the Hippo pathway. We identify a total of 85 loss-of-function (LOF) missense mutations for Hippo pathway genes and elucidate their underlying mechanisms. Interestingly, we reveal zinc-finger domain as an integral structure for MOB1 function, whose LOF mutations in head and neck cancer promote tumor growth. Moreover, the schwannoma/meningioma-derived NF2 LOF mutations not only inhibit its tumor suppressive function in the Hippo pathway, but also gain an oncogenic role for NF2 by activating the VANGL-JNK pathway. Collectively, our study not only offers a rich somatic mutation resource for investigating the Hippo pathway in human cancers, but also provides a molecular basis for Hippo-based cancer therapy.

The Hippo pathway is a key regulator of development, regeneration, organ size, tissue homeostasis, and cancer[1–4]. It senses and transduces a wide range of growth-related signals to restrict proliferation and stimulate apoptosis, two events that are frequently lost during tumorigenesis. However, precisely how the Hippo pathway becomes dysregulated in tumorigenesis and how such alterations affect the Hippo pathway to drive human cancer development have not been fully understood.

The mammalian Hippo pathway comprises two STE20-like kinases MST1 and MST2 (MST1/2), two AGC family of kinases LATS1 and LATS2 (LATS1/2), and their adaptor proteins SAV1 and MOB1A and MOB1B (MOB1A/B), respectively. NF2 is a membrane-bound Hippo component required for LATS1/2 membrane translocation and activation[5]. As additional core kinase components, MAP4K1-7 act in parallel to MST1/2 to phosphorylate and activate LATS1/2[6–10]. Once activated, LATS1/2 phosphorylate two transcriptional co-activators, YAP and TAZ,

[1]Department of Pathophysiology, TaiKang Medical School (School of Basic Medical Sciences), Wuhan University, Wuhan, Hubei, China. [2]TaiKang Center for Life and Medical Sciences, Wuhan University, Wuhan, Hubei, China. [3]Department of Developmental and Cell Biology, University of California, Irvine, Irvine, CA, USA. [4]Chemical and Materials Physics Graduate Program, University of California, Irvine, Irvine, CA, USA. [5]Department of Chemistry and Shenzhen Grubbs Institute, Guangdong Provincial Key Laboratory of Catalysis, Southern University of Science and Technology, Shenzhen, Guangdong, China. [6]Department of Chemistry, University of California, Irvine, Irvine, CA, USA. [7]Department of Molecular and Cellular Pharmacology and Sylvester Comprehensive Cancer Center, University of Miami Miller School of Medicine, Miami, FL, USA. [8]School of Life Sciences, Westlake University, Hangzhou, Zhejiang, China. [9]Cryo-EM Center, Southern University of Science and Technology, Shenzhen, Guangdong, China. [10]Department of Molecular Biology and Biochemistry, University of California, Irvine, Irvine, CA, USA. [11]Department of Chemical and Biomolecular Engineering, University of California, Irvine, Irvine, CA, USA. [12]Department of Materials Science and Engineering, University of California, Irvine, Irvine, CA, USA. [13]Department of Biomedical Engineering, University of California, Irvine, Irvine, CA, USA. [14]These authors contributed equally: Han Han, Zhen Huang, Congsheng Xu. ✉e-mail: hanhan@whu.edu.cn; qirx@sustech.edu.cn; rluo@uci.edu; wenqiw6@uci.edu

resulting in their cytoplasmic retention and degradation. When the Hippo pathway is inactivated, unphosphorylated YAP and TAZ are translocated into the nucleus, where they bind transcription factors like TEAD1-4 to initiate transcription of genes involved in numerous growth-related events.

Genetic depletion of the Hippo pathway components or transgenic overexpression of its downstream effector YAP displays increased organ size and eventually tumor growth in mice[1–4]. YAP and TAZ are highly expressed and activated in various types of human cancers, where they promote cell transformation, tumor growth, metastasis, relapse, chemo-resistance, and cancer stem cell maintenance[11,12]. The Cancer Genome Atlas (TCGA) study also reveals the Hippo pathway as one of the commonly altered signaling pathways involved in cancer development[13]. These facts pinpoint a crucial role of the Hippo pathway in cancer, inspiring both basic and translational research interests in this pathway over the past decades.

Despite the rapid progress in the field, our understanding of the Hippo pathway in human cancers remains incomplete. Although several bioinformatic studies have profiled the Hippo pathway in human cancer genome[13–16], they provided only a simple glimpse into the Hippo pathway dysregulation in human cancers, lacking functional validation and mechanistic insights. Additionally, our current Hippo-related cancer research largely depends on knockout/knockdown- and overexpression-based approaches, while the expression of major Hippo pathway components is infrequently changed in major types of human cancers. Characterizing Hippo pathway somatic mutations may help address this issue, while the related research has been scarce.

In this study, we analyzed the Hippo somatic mutations derived from the human cancer genome and functionally characterized ~1000 missense mutations for the major Hippo pathway components. Through a YAP-based screen study conducted in the Hippo component knockout (KO) cells, we identified a total of 85 Hippo loss-of-function (LOF) mutations that can inhibit the Hippo pathway and 945 neutral mutations that failed to do so. We experimentally validated the Hippo LOF mutations and elucidated the mechanisms by which these mutations targeted the Hippo pathway. Through these studies, we revealed the zinc-finger (ZNF) domain as an essential region for MOB1 protein folding and function, whose head and neck cancer-associated LOF mutations inhibited the Hippo pathway and promoted head and neck tumor growth. In addition, we showed that the meningioma/schwannoma-derived NF2 LOF mutations not only compromised its tumor suppressive function in the Hippo pathway, but also gained an oncogenic role for NF2 by activating the VANGL-JNK pathway. Taken together, our study functionally illustrates the Hippo somatic mutations in the human cancer genome, providing mechanistic insights into the Hippo pathway dysregulation in human cancers from a pathological perspective.

## Results

### Overview of the Hippo signaling alterations in the human cancer genome

To study Hippo signaling dysregulation in human cancers, we analyzed the somatic alterations of Hippo pathway components, effectors YAP/TAZ, and transcription factors TEAD1-4 (hereafter referred to as Hippo signaling) in The Cancer Genome Atlas (TCGA). Among a total of 11,706 cancer patient samples across 32 cancer types, 4134 samples carried the altered Hippo signaling genes, making its total alteration rate up to 35.3% (Fig. 1a and Supplementary Data 1). Some cancer patients even carried multiple altered Hippo signaling genes (Fig. 1b and Supplementary Data 1). The cancer samples with the altered Hippo signaling genes were featured with high histologic grades (Fig. 1c and Supplementary Data 1), high tumor stages (Fig. 1d and Supplementary Data 1), and high disease stages (Fig. 1e and Supplementary Data 1). As for each Hippo signaling gene, cancer patient samples with Hippo signaling gene alterations mostly exhibited high histologic grades

(Supplementary Fig. 1a and Supplementary Data 1); however, this trend was not observed in studies of tumor stage (Supplementary Fig. 1b and Supplementary Data 1) or disease stage (Supplementary Fig. 1c and Supplementary Data 1).

Consistent with a previous study[14], the somatic alteration rate of each Hippo signaling gene was less than 10% (Fig. 1f and Supplementary Data 1), suggesting that Hippo signaling dysregulation could be caused by its concomitantly altered genes (Fig. 1b) or deregulated periphery regulators. MST2 and MAP4K7 are two Hippo pathway components amplified in human cancers (Fig. 1f). Although they are generally considered as tumor suppressors due to their roles in the Hippo pathway, MST2 and MAP4K7 also have YAP/TAZ-independent functions that drive cancer development[17–19], suggesting the complex roles of altered Hippo signaling genes in human cancer development.

Among the 32 cancer types in TCGA, the altered Hippo signaling genes were observed in different types of human cancers and highly enriched in lymphoid neoplasm diffuse large B-cell/lymphoma (DLBC), esophageal carcinoma (ESCA), skin cutaneous melanoma (SKCM), stomach adenocarcinoma (STAD), uterine corpus endometrial carcinoma (UCEC), and uterine carcinosarcoma (UCS) (Fig. 1g and Supplementary Data 1). Regarding clinical relevance, bladder urothelial carcinoma (BLCA) patient samples with Hippo signaling alterations exhibited high histological grades (Supplementary Fig. 2a and Supplementary Data 1). Prostate adenocarcinoma (PRAD) patient samples with altered Hippo signaling genes were significantly correlated with advanced tumor stages (Supplementary Fig. 2b and Supplementary Data 1). Kidney chromophobe (KICH) and kidney renal papillary cell carcinoma (KIRP) showed significant correlations between altered Hippo signaling genes and increased disease stage (Supplementary Fig. 2c and Supplementary Data 1). In contrast, colon adenocarcinoma (COAD) cancer patient samples with Hippo signaling alterations exhibited decreased disease stages (Supplementary Fig. 2c and Supplementary Data 1). Additionally, cancer patients with Hippo signaling alterations showed poor overall survival rates as compared to those with normal Hippo signaling genes (Supplementary Fig. 2d). Specifically, ESCA, KICH, kidney renal clear cell carcinoma (KIRC), liver hepatocellular carcinoma (LIHC), and PRAD patients with altered Hippo signaling genes exhibited poor overall survival (Supplementary Fig. 2e); however, Hippo signaling gene alterations indicate better overall survival rates for BLCA and glioblastoma (GBM) patients (Supplementary Fig. 2e). These findings suggest the complex and context-dependent nature of Hippo signaling alterations in different cancers.

As for the somatic mutations, nonsense mutation was mostly found in Hippo pathway genes *NF2*, *SAV1*, *LATS1* and *MOB1B* (Fig. 1h and Supplementary Data 2). A significant proportion of cancer patients carried the fusion- and frame-related mutations in *YAP1* and *TEAD2*, respectively (Fig. 1h and Supplementary Data 2). Notably, non-silent missense mutation was the dominant somatic mutation type for all the Hippo pathway genes (Fig. 1h and Supplementary Data 2). This type of somatic mutation is mysterious because it only results in substitution of one amino acid to be a different one but could largely change protein function. Therefore, we were interested in the Hippo pathway missense mutations and further characterized them in this study.

### Functional annotation of the Hippo pathway missense mutations

To achieve this goal, we turned to the Hippo pathway component knockout (KO) cells, where YAP is dominantly localized in the nucleus of the LATS1/2 double KO (DKO), MOB1A/B DKO, MST/MAP4K-8KO, and NF2 KO cells (Supplementary Fig. 3a). Importantly, the YAP cytoplasmic localization can be fully rescued when wild-type Hippo genes were reconstituted into their corresponding KO cells, while their inactive mutants (e.g., the kinase dead mutants of LATS1, MST1, MST2)

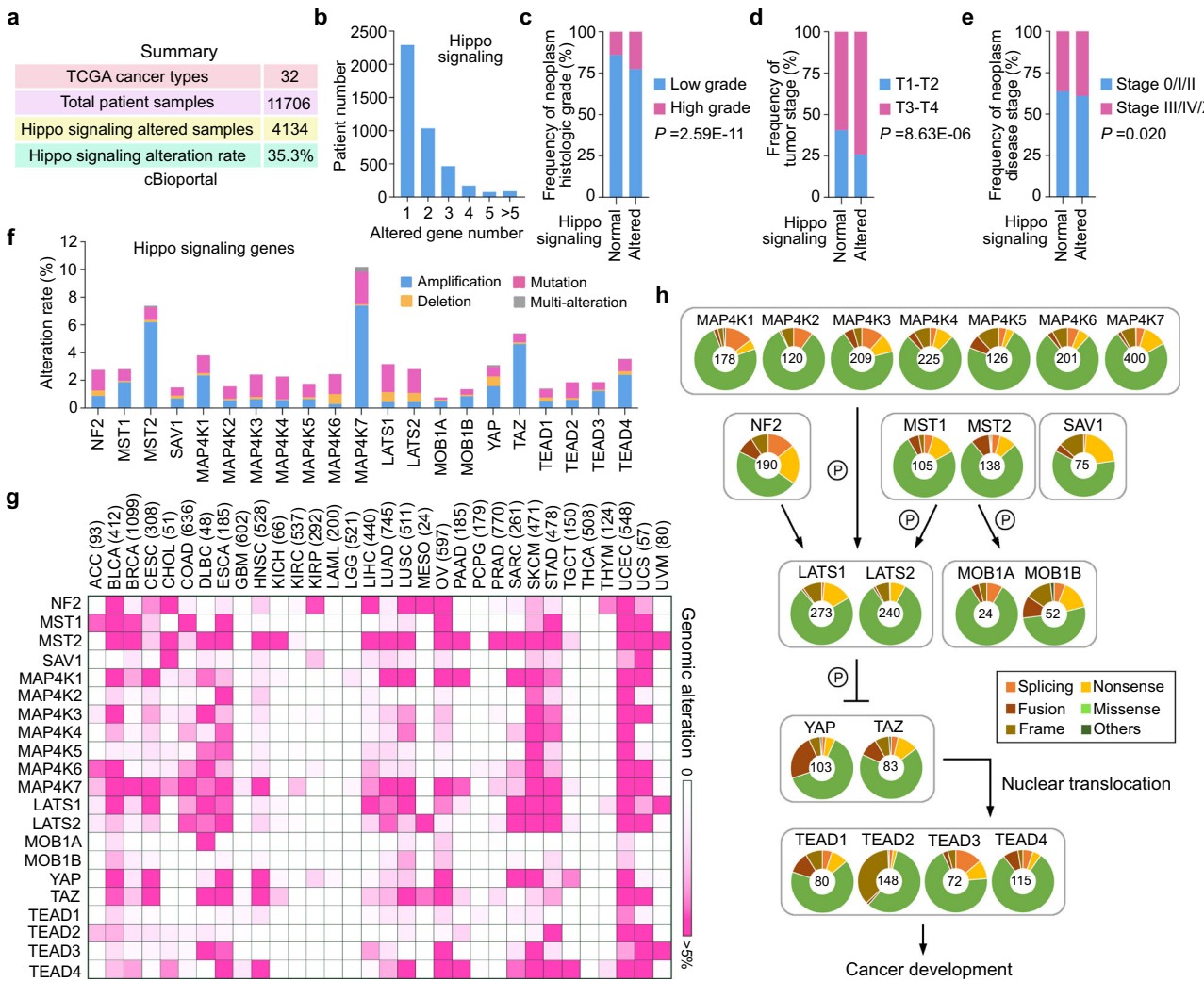

**Fig. 1 | Analysis of Hippo signaling gene alterations in the human cancer genome. a** Summary of Hippo signaling gene alterations in TCGA. Information of the TCGA patient samples was obtained from cBioportal. Hippo signaling genes refer to the Hippo pathway core components, effectors YAP and TAZ, and transcription factors TEAD1-4. **b** Around 1500 cancer patient samples carry multiple altered Hippo signaling genes. **c–e** Histological analysis of the cancer patient samples with the altered Hippo signaling genes. Tumor samples were analyzed for their neoplasm histologic grade (**c**), tumor stage (**d**), and neoplasm disease stage (**e**) using chi-squared test. **f** Summary of the alteration types for Hippo signaling genes. Hippo signaling genes were analyzed for their amplification, deletion, mutation, and multi-alterations (i.e., mutation occurs together with either amplification or deletion) in TCGA patient samples. **g** Overview of the Hippo signaling gene alterations across 32 cancer types in TCGA. Total patient sample number for each cancer type was indicated. **h** Illustration of the Hippo signaling gene somatic mutations in TCGA. The number of patient samples carrying the somatic mutations was indicated for each Hippo signaling gene. Source data are provided as a Source Data file.

failed to do so (Supplementary Fig. 3b, c). These data demonstrate that these KO cells can serve as a robust tool to functionally annotate Hippo pathway missense mutations.

Next, we generated ~1000 Hippo missense mutants for 9 major Hippo pathway components (LATS1, LATS2, MOB1A, MOB1B, MST1, MST2, MAP4K2, MAP4K3, NF2) based on TCGA and expressed them individually in their corresponding KO cells (Fig. 2a and Supplementary Data 3). If the missense mutants failed to rescue YAP cytoplasmic localization in the KO cells, we annotated these mutations as the "loss-of-function (LOF)" mutations. In contrast, if the missense mutants were still able to translocate YAP into the cytoplasm, we referred to these mutations as "neutral" mutations (Fig. 2a). Through the screen, we identified a total of 85 LOF missense mutations for the Hippo pathway genes (Fig. 2b and Supplementary Data 3), which were carried by 95 cancer patients (Fig. 2d and Supplementary Data 3) and distributed in different types of human cancers (Fig. 2f and Supplementary Data 3). These Hippo LOF mutations were significantly enriched in BLCA, mesothelioma (MESO), SKCM and UCEC (Supplementary Data 3), suggesting their critical roles in development and/or progression of

these cancer types. As for Hippo pathway components, multiple LOF mutations were revealed for Hippo pathway kinases in different types of human cancers (Fig. 2g); only one LOF mutation was uncovered for MOB1A and MOB1B each in head and neck squamous cell carcinoma (HNSC) (Fig. 2f); and the NF2 LOF mutations were identified in meningioma and schwannoma patients (Fig. 2f) consistent with previous findings[20,21]. Taken together, our functional profiling reveals a group of LOF missense mutations for the major Hippo pathway components.

Our study also uncovered a total of 945 Hippo pathway neutral mutations from 1111 cancer patients (Fig. 2c, e and Supplementary Data 3). We randomly selected approximately 20 neutral mutations for the Hippo kinases MST1, MST2 and MAP4K3, and expressed them individually in the LATS1/2 DKO cells. In contrast to LATS1 and LATS2 (Supplementary Fig. 3b), neither the wild-type MST/MAP4K proteins nor their neutral mutants (Supplementary Fig. 4a-c) were able to rescue YAP's cytoplasmic localization in the LATS1/2 DKO cells. These results highlight the essential role of LATS1/2 in mediating the inhibitory effect of MST/MAP4K neutral mutations on YAP.

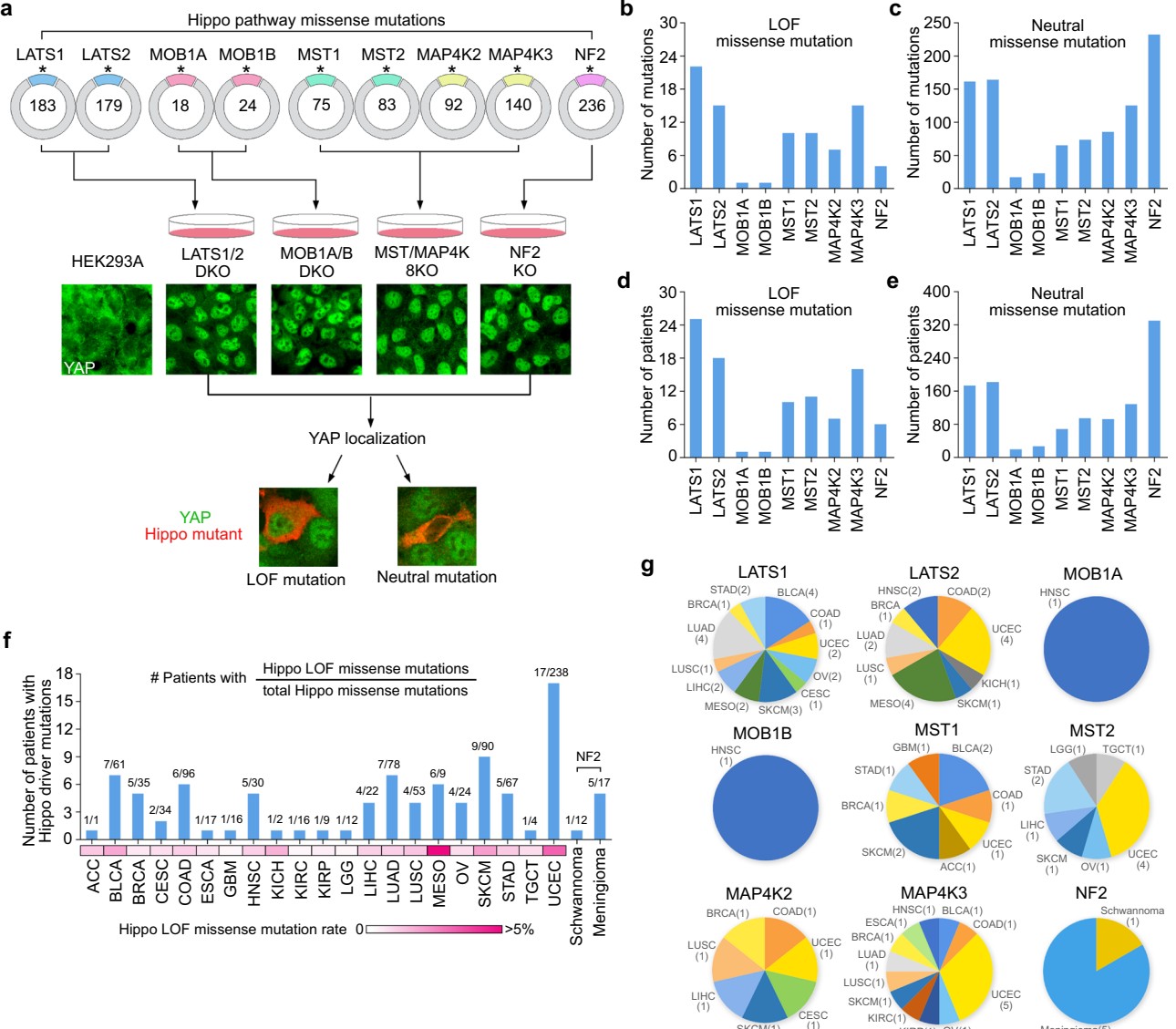

**Fig. 2 | Functional annotation of the Hippo pathway somatic mutations.**
**a** Illustration of the workflow for characterizing Hippo pathway missense muta-
tions. The TCGA-documented Hippo pathway missense mutants were generated
and reconstituted into their corresponding KO cells. The total missense mutation
number for each Hippo pathway gene was indicated. **b**, **c** Summary of the identified
LOF (**b**) and neutral (**c**) missense mutations for each Hippo pathway gene.
**d**, **e** Summary of the cancer patients carrying the LOF (**d**) and neutral (**e**) missense
mutations for each Hippo pathway gene. **f** Summary of the cancer patients carrying

the Hippo LOF missense mutations across different cancer types. For each cancer
type, the numbers of patients carrying the LOF and total missense mutations of the
Hippo pathway genes were indicated. The Hippo LOF missense mutation rate for
each cancer type was shown as a heatmap. The information of NF2 missense
mutations in schwannoma and meningioma was obtained from COSMIC.
**g** Summary of the LOF missense mutations identified for each Hippo pathway gene.
The cancer type and mutation number were indicated. Source data are provided as
a Source Data file.

## Analysis of the Hippo oncogenic alterations in the human cancer genome

Next, we combined the identified Hippo LOF missense mutations with
the TCGA-documented oncogenic alterations (i.e., deletion, nonsense
mutation, gene fusion, frame mutation, splicing mutation, translation
start site mutation) that can inactivate the Hippo pathway components
as the Hippo pathway oncogenic alterations. As for YAP/TAZ and
TEAD1-4, their oncogenic alterations were obtained based on TCGA
and a previous missense mutation screen[14].

As shown in Supplementary Fig. 5a and Supplementary Data 4, the
Hippo pathway genes were affected by different types of oncogenic
alterations, while genomic amplification (AMP) was the dominant
alteration type for YAP/TAZ and TEAD1-4. We identified a total of 1251
and 1313 cancer patient samples carrying the oncogenic alterations for
the Hippo pathway genes and YAP/TAZ/TEAD1-4, respectively

(Supplementary Fig. 5a and Supplementary Data 4). This made the rate
of affected patient samples in TCGA up to 8% for the Hippo pathway
genes, 10% for YAP/TAZ/TEAD1-4, and 17% for their integrated Hippo
signaling (Supplementary Fig. 5b and Supplementary Data 4). These
rates were less than that of tumor suppressor gene *TP53* (36%) but
comparable to or even higher than that of tumor suppressor gene
*PTEN* (11%) and oncogenes *PI3KCA* (17%), *KRAS* (10%) and *HRAS* (1%)
based on TCGA (Supplementary Fig. 5a, b and Supplementary Data 4).

Collectively, these data present an overview of the Hippo signal-
ing oncogenic alterations, consistently showing that the dysregulated
Hippo signaling is a frequent event in human cancers.

## Characterization of the LATS1/2 LOF missense mutations

We identified a total of 22 LOF missense mutations for LATS1 (Fig. 2b
and Supplementary Data 3), among which 5 mutations were in its

MOB1-binding domain (MBD) and the rest 17 were within its kinase domain (Fig. 3a). As for the 15 LOF mutations identified for LATS2 (Fig. 2b), one of them was in its MBD and the rest 14 mutations were in its kinase domain (Fig. 3b). To validate them, we subjected each

purified LATS1/2 LOF mutant protein to in vitro kinase assay using the bacterially purified GST-YAP as substrate (Fig. 3c). As shown in Fig. 3d–g, all the MBD- and kinase domain-associated LOF mutations abolished the LATS1/2-mediated YAP S127 phosphorylation. Moreover,

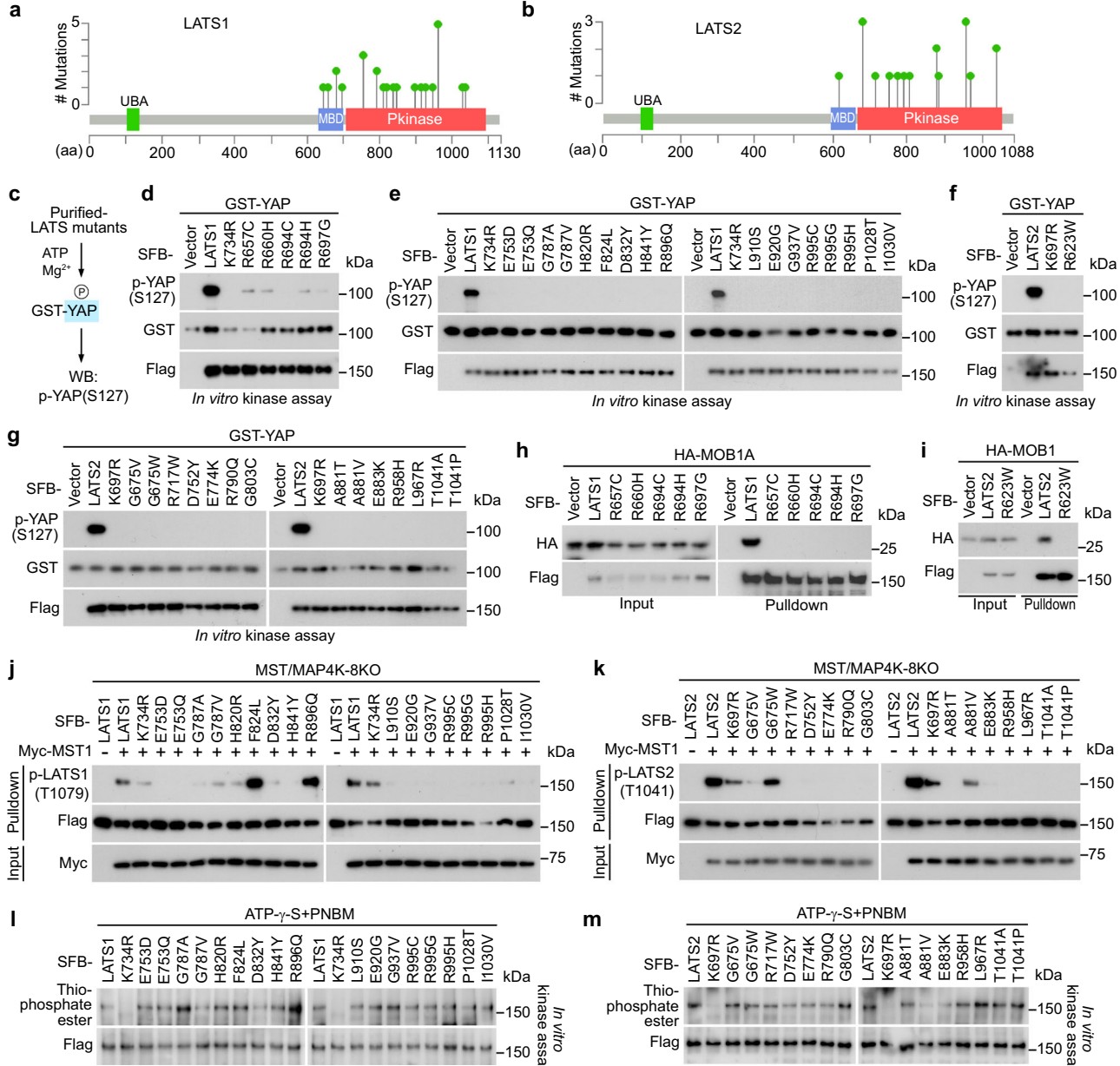

**Fig. 3 | Characterization of the LATS1/2 LOF missense mutations. a**, **b** Illustration of the protein domains and the identified LOF missense mutations for LATS1/2. **c** Illustration of the in vitro kinase assay used for validating the LATS1/2 LOF missense mutations. The purified LATS1/2 LOF missense mutants were subjected to in vitro kinase assay using bacterially purified GST-YAP as substrate. **d**, **e** The LATS1 LOF missense mutations inhibit LATS1 activity. The SFB-tagged LATS1 MOB1-binding domain (MBD)-associated LOF mutants (**d**) and its kinase domain-associated LOF mutants (**e**) were expressed in HEK293T cells, purified using S protein beads, washed thoroughly with high-salt buffer containing 250 mM NaCl, and subjected to in vitro kinase assay using bacterially purified GST-YAP protein as substrate. A representative blot of two independent experiments is shown. **f**, **g** The LATS2 LOF missense mutations inhibit LATS2 activity. The SFB-tagged LATS2 MOB1-binding domain (MBD)-associated LOF mutants (**f**) and its kinase domain-associated LOF mutants (**g**) were expressed in HEK293T cells, purified using S protein beads, washed thoroughly with high-salt buffer containing 250 mM NaCl, and subjected to in vitro kinase assay using bacterially purified GST-YAP protein as

substrate. A representative blot of two independent experiments is shown. **h**, **i** The LATS1/2 MBD-associated LOF missense mutations disrupt the interaction of LATS1/2 with MOB1. HEK293T cells were transfected with HA-MOB1 and the indicated SFB-tagged LOF mutants of LATS1 (**h**) and LATS2 (**i**) and subjected to pulldown assay using S protein beads. A representative blot of two independent experiments is shown. **j**, **k** The LATS1/2 kinase domain-associated LOF missense mutations inhibit their hydrophobic motif (HM) phosphorylation by MST1. The MST/MAP4K-8KO HEK293A cells were transfected with Myc-MST1 and the indicated SFB-tagged LOF mutants of LATS1 (**j**) and LATS2 (**k**) and subjected to pulldown assay. A representative blot of two independent experiments is shown. **l**, **m** The LATS1/2 kinase domain-associated LOF missense mutations do not affect LATS1/2 ATP-binding abilities. The LATS1/2-DKO HEK293A cells were transfected with the indicated SFB-tagged LOF mutants of LATS1 (**l**) and LATS2 (**m**), purified using S protein beads, washed thoroughly with high-salt buffer containing 250 mM NaCl, and subjected to γ-ATP-based in vitro kinase assay. A representative blot of two independent experiments is shown. Source data are provided as a Source Data file.

these LATS1/2 LOF mutants all failed to rescue YAP S127 phosphorylation when stably reconstituted into the LATS1/2 DKO cells (Supplementary Fig. 6a, b). These data confirm the inhibitory roles of LATS1/2 LOF mutations in regulating LATS1/2 activities.

As for the MBD-associated LOF mutations, they all disrupted the interaction of LATS1/2 with MOB1 (Fig. 3h, i). This can be explained by the facts that LATS1 R694 and R697 residues were required for the contact with MOB1 while LATS1 R657 and R660 residues and LATS2 R623 residues were involved in the LATS1/2-MBD structure maintenance[22]. As for the LATS1/2 kinase domain-associated LOF mutations, we first examined whether they could affect the LATS1/2-hydrophobic motif (HM) phosphorylation by MST1. Among the 17 LATS1 kinase domain-associated LOF mutations (Fig. 3a), 15 mutations largely interfered with the MST1-induced LATS1-HM phosphorylation at T1079 (Fig. 3j), indicating these LATS1 mutants are poor substrates for MST1. In contrast, the rest two LOF mutations F824L and R896Q even further enhanced the MST1-induced LATS1-HM phosphorylation at T1079 (Fig. 3j). We speculate these two LATS1 mutants may have a reduced intrinsic kinase activity or lost ability to phosphorylate YAP even though they can be effectively phosphorylated by MST1. Moreover, the LATS2 kinase domain-associated 14 LOF mutations including two directly occurred at its T1041 residue (i.e., T1041A, T1041P) (Fig. 3b) all inhibited MST1-induced its HM phosphorylation at T1041 (Fig. 3k). As a second readout, the ATP-binding ability of LATS1/2 was examined through a γ-ATP-based in vitro kinase assay. As shown in Fig. 3l, m, in contrast to their ATP-binding mutation controls (i.e., LATS1 K734R, LATS2 K697R), the LATS1/2 kinase domain-associated LOF mutations did not affect their ATP-binding abilities.

To gain a structural insight into the LATS1/2 kinase domain-associated LOF mutations, we performed a molecular dynamics (MD) analysis for these mutations using the AlphaFold2 predicted structures for LATS1 (Supplementary Fig. 7a) and LATS2 (Supplementary Fig. 8a). As shown in Supplementary Fig. 7b, all the identified LOF mutations induced a confirmational change for LATS1 kinase domain. Our root-mean-square deviation (RMSD) analysis further revealed that such structural changes of LATS1 kinase domain majorly happened to its helix group (Supplementary Fig. 7c) but not its HM (Supplementary Fig. 7d) or β-sheet region (Supplementary Fig. 7e). Moreover, all the LATS1 kinase domain-associated LOF mutations largely affected the fluctuation of the residues surrounding its HM phosphorylation site T1079 (Supplementary Fig. 7f), providing potential mechanisms for the altered LATS1 T1079 phosphorylation as caused by its LOF mutations (Fig. 3j). As a control, the residues near the LATS1 ATP-binding site K734 showed only a mild fluctuation change (Supplementary Fig. 7f), which was consistent with our experimental data (Fig. 3l). The LATS2 kinase domain-associated LOF mutations also changed its kinase domain structure (Supplementary Fig. 8b), which majorly occurred at its helix group (Supplementary Fig. 8c) and HM (Supplementary Fig. 8d) but not the β-sheet region (Supplementary Fig. 8e). Similarly, almost all the LATS2 LOF mutations significantly affected the fluctuation of the residues surrounding its HM phosphorylation site T1041, while this was not the case for the residues near its ATP-binding site K697 (Supplementary Fig. 8f). These findings were consistent with our experimental data regarding the LOF mutations-induced effects on LATS2-HM T1041 phosphorylation (Fig. 3k) and its ATP-binding ability (Fig. 3m).

Taken together, these results not only validate the identified LATS1/2 LOF mutations, but also provide mechanistic insights into their inhibitory effects on LATS1/2.

### Elucidation of the LOF missense mutations for MST1/2
A total of 10 LOF missense mutations were identified for MST1 and MST2 each (Fig. 2b and Supplementary Data 3), and they were all located in their kinase domains (Fig. 4a, b). We subjected each purified MST1/2 LOF mutant protein to in vitro kinase assay using the bacterially purified MBP-LATS1-HM-containing protein C3 (i.e., LATS1-C3)[23] as substrate (Fig. 4c). Indeed, all the identified MST1/2 LOF mutations dramatically diminished their abilities to phosphorylate LATS1-C3 at T1079 (Fig. 4d, e). In addition, all the MST1/2 LOF mutants failed to rescue MOB1 T35 phosphorylation in the MST/MAP4K-8KO cells (Fig. 4f, g and Supplementary Fig. 6c). These results confirm the LOF mutations-caused negative effects on MST1/2 both in vitro and in vivo.

Next, we examined the autophosphorylation and ATP-binding abilities of the MST1/2 LOF mutants. First, the MST1/2 LOF mutants were co-expressed with wild-type MST1/2 in the MST/MAP4K-8KO cells and purified to examine their autophosphorylation at T183/T180. As shown in Fig. 4h, 5 LOF mutations I155T, R181Q, E219K, P223S, and P229Q prevented MST1 from being phosphorylated at T183 by wild-type MST1, while the other 5 LOF mutations did not affect the intermolecular autophosphorylation. Moreover, the identified 10 LOF mutations of MST2 all inhibited its T180 phosphorylation as induced by wild-type MST2 (Fig. 4i). Regarding the ATP-binding ability, except for the D97V and W99L mutations, the rest 8 MST1 LOF mutations all interfered with the ATP-binding ability of MST1 (Fig. 4j). As for the MST2 LOF mutations, they all largely abolished its ATP-binding ability (Fig. 4k). These data show that most of the identified MST1/2 LOF mutations can affect at least one of these two key events required for MST1/2 activation. As for the MST1 D97V and W99L mutations that did not interfere with either its autophosphorylation or ATP-binding (Fig. 4h, j), they may affect MST1 to recognize its substrates (e.g., LATS1 and MOB1) or through some yet-to-be-identified mechanisms.

We performed similar MD analyses using the available protein structures for MST1 (Supplementary Fig. 9a) and MST2 (Supplementary Fig. 10a). As for MST1, all the identified LOF mutations caused a conformational change for the MST1 kinase domain (Supplementary Fig. 9b). Specifically, the structures of MST1 activation loop (Supplementary Fig. 9c) and helix group (Supplementary Fig. 9d) were significantly altered by most of its LOF mutations as compared to that of its β-sheet region (Supplementary Fig. 9e). These findings were experimentally confirmed via a circular dichroism (CD) analysis (Fig. 4l, m). In addition, the MST1 LOF mutations altered the residue fluctuation across the MST1 kinase domain, including the regions surrounding its autophosphorylation site T183 and ATP-binding site K59 (Supplementary Fig. 9f). Similarly, the MST2 kinase domain structure was also changed by its LOF mutations (Supplementary Fig. 10b). Specifically, most of the MST2 LOF mutations induced a conformational change for the activation loop (Supplementary Fig. 10c), helix group (Supplementary Fig. 10d) and β-sheet region (Supplementary Fig. 10e) of MST2 kinase domain. These results were consistent with our CD results showing that most of the MST2 LOF mutations caused a structural change for the bacterially purified MST2 kinase domain protein (Fig. 4n, o). We also observed a fluctuation change for the MST2 kinase domain residues as caused by its LOF mutations, which majorly occurred at the regions nearby its autophosphorylation site T180 (except for the T180I and R266I mutations) and ATP-binding site K56 (Supplementary Fig. 10f). Collectively, these MD data show that the MST1/2 kinase domain structures are significantly affected by their LOF mutations, providing potential mechanisms for their inhibitory effects on MST1/2.

### Investigation of the LOF missense mutations for MAP4K2/3
We identified a total of 7 and 15 LOF missense mutations for MAP4K2 and MAP4K3, respectively (Fig. 2b and Supplementary Data 3), which were all in their kinase domains (Fig. 5a, b). To validate them, we purified MAP4K2/3 LOF mutant proteins and subjected them to in vitro kinase assay using bacterially purified MBP-LATS1-C3 as substrate (Fig. 5c). As shown in Fig. 5d, e, the identified LOF mutations all abolished the abilities of MAP4K2/3 to phosphorylate LATS1-C3 at T1079.

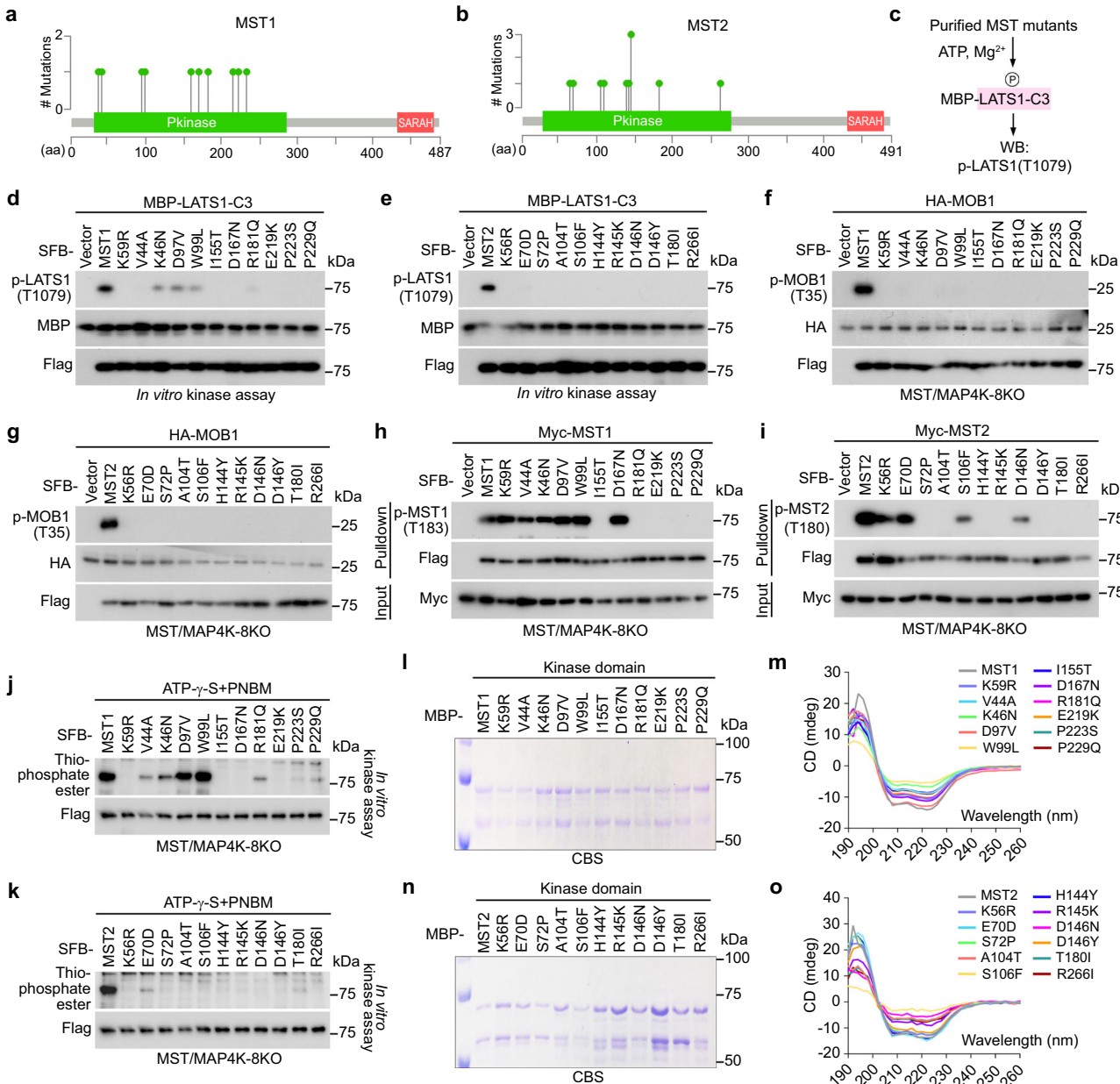

**Fig. 4 | Analysis of the MST1/2 LOF missense mutations. a, b** Illustration of the protein domains and the identified LOF missense mutations for MST1/2. **c** Illustration of the in vitro kinase assay used for validating the MST1/2 LOF missense mutations. The purified MST1/2 LOF missense mutants were subjected to in vitro kinase assay using bacterially purified MBP-LATS1-C3 as substrate. **d, e** The MST1/2 LOF missense mutations abolish their kinase activities in vitro. The SFB-tagged LOF mutants of MST1 (**d**) and MST2 (**e**) were expressed in HEK293T cells, purified using S protein beads, washed thoroughly with high-salt buffer containing 250 mM NaCl, and subjected to in vitro kinase assay using bacterially purified MBP-LATS1-C3 protein as substrate. A representative blot of two independent experiments is shown. **f, g** The MST1/2 LOF missense mutations abolish their kinase activities in vivo. The MST/MAP4K-8KO HEK293A cells were transfected with HA-MOB1 and the indicated SFB-tagged LOF mutants of MST1 (**f**) and MST2 (**g**) and subjected to Western blot. A representative blot of two independent experiments is shown. **h** Characterization of the MST1 LOF missense mutations-induced effects on its autophosphorylation ability. The MST/MAP4K-8KO HEK293A cells were transfected with Myc-MST1 and the indicated SFB-MST1 LOF mutants and subjected to pulldown assay. A representative blot of two independent experiments is shown.

**i** Characterization of the MST2 LOF missense mutations-induced effects on its autophosphorylation ability. The MST/MAP4K-8KO HEK293A cells were transfected with Myc-MST2 and the indicated SFB-MST2 LOF mutants and subjected to pulldown assay. A representative blot of two independent experiments is shown. **j, k** Analysis of the MST1/2 LOF missense mutations-induced effects on their ATP-binding abilities. The MST/MAP4K-8KO HEK293A cells were transfected with the indicated SFB-tagged LOF mutants of MST1 (**j**) and MST2 (**k**), purified using S protein beads, washed thoroughly with high-salt buffer containing 250 mM NaCl, and subjected to γ-ATP-based in vitro kinase assay. A representative blot of two independent experiments is shown. **l, m** The MST1 LOF missense mutations induce a conformational change of its kinase domain. The indicated MBP-tagged MST1-kinase domain proteins were purified from bacteria (**l**) and subjected to circular dichroism analysis (**m**). CBS, Coomassie blue staining. **n, o** The MST2 LOF missense mutations induce a conformational change of its kinase domain. The indicated MBP-tagged MST2-kinase domain proteins were purified from bacteria (**n**) and subjected to circular dichroism analysis (**o**). CBS, Coomassie blue staining. Source data are provided as a Source Data file.

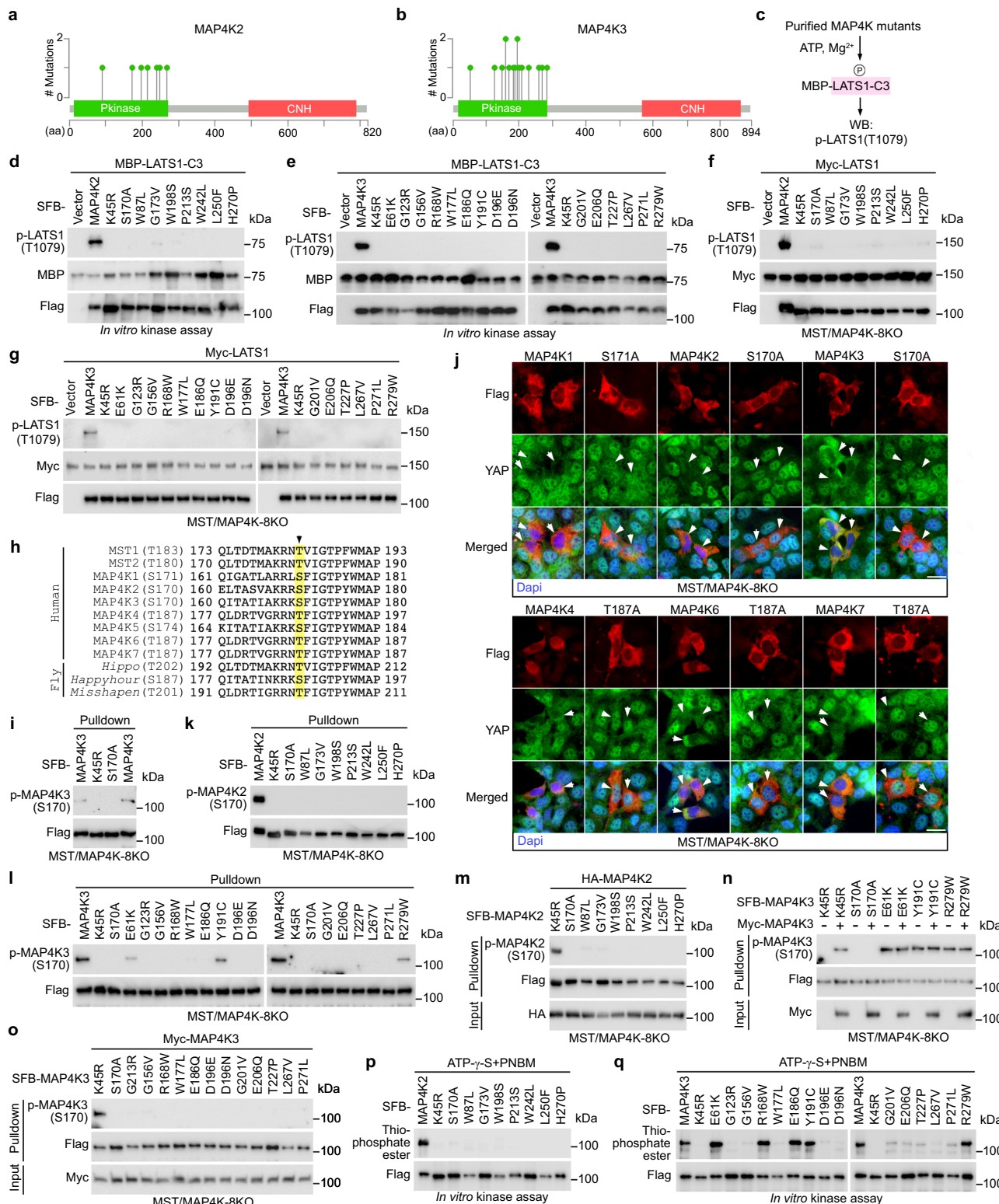

Moreover, the MAP4K2/3 LOF mutants all failed to rescue LATS1 T1079 phosphorylation when reconstituted in the MST/MAP4K-8KO cells (Fig. 5f, g). These results confirm the inhibitory effects of the MAP4K2/3 LOF mutations.

To elucidate the mechanism, first, we focused on their autophosphorylation event, because MAP4Ks belong to the STE20-like kinase family and its members like MST1/2 are known to activate themselves via autophosphorylation[24–28]. Interestingly, protein sequence alignment revealed a potential autophosphorylation site for all the MAP4K-family kinases, which is in their kinase domains and conserved in their *Drosophila* orthologs *Happyhour* and *Misshapen* (Fig. 5h). Indeed, our recent study has confirmed this prediction (Fig. 5h) for MAP4K2[29]. MAP4K3 was also phosphorylated at the predicted S170 site (Fig. 5h, i), while this was not the case for its kinase dead mutant (i.e., K45R) (Fig. 5i). Mutating the predicted autophosphorylation site (Fig. 5h) to Ala abolished the abilities of the MAP4K-family kinases to rescue YAP cytoplasmic localization in the MST/MAP4K-8KO cells (Fig. 5j). These data suggest that the predicted autophosphorylation site is essential

**Fig. 5 | Elucidation of the MAP4K2/3 LOF missense mutations. a, b** Illustration of the protein domains and the identified LOF missense mutations for MAP4K2/3. **c** Illustration of the in vitro kinase assay used for validating the MAP4K2/3 LOF missense mutations. **d, e** The MAP4K2/3 LOF missense mutations target their kinase activities in vitro. The SFB-tagged LOF mutants of MAP4K2 (**d**) and MAP4K3 (**e**) were expressed in HEK293T cells, purified using S protein beads, washed with high-salt buffer containing 250 mM NaCl, and subjected to in vitro kinase assay. A representative blot of two independent experiments is shown. **f, g** The MAP4K2/3 LOF missense mutations abolish their kinase activities in vivo. The MST/MAP4K-8KO HEK293A cells were transfected with Myc-LATS1 and the indicated SFB-tagged LOF mutants of MAP4K2 (**f**) and MAP4K3 (**g**) and subjected to Western blot. A representative blot of two independent experiments is shown. **h** Protein sequence alignment reveals a potential autophosphorylation site for MAP4K-family kinases. The predicted autophosphorylation site for MAP4Ks was highlighted. **i** MAP4K3 activity is required for its S170 phosphorylation. The MST/MAP4K-8KO cells were transfected with the indicated SFB-MAP4K3 constructs and subjected to pulldown assay. A representative blot of two independent experiments is shown. **j** Mutation of the predicted autophosphorylation site inhibits MAP4Ks. The MST/MAP4K-8KO HEK293A cells were transfected with the indicated constructs and subjected to immunofluorescent staining. Nucleus was visualized by Dapi. Scale bar, 20 μm.

Arrows showed the cells expressing the indicated constructs. **k, l** The MAP4K2/3 autophosphorylation is dramatically inhibited by their LOF missense mutations. The MST/MAP4K-8KO HEK293A cells were transfected with the indicated SFB-tagged LOF mutants of MAP4K2 (**k**) and MAP4K3 (**l**) and subjected to pulldown assay. A representative blot of two independent experiments is shown. **m** Characterization of the MAP4K2 LOF missense mutations-induced effects on its autophosphorylation ability. The MST/MAP4K-8KO HEK293A cells were transfected with HA-MAP4K2 and the indicated SFB-MAP4K2 LOF mutants and subjected to pulldown assay. A representative blot of two independent experiments is shown. **n, o** Characterization of the MAP4K3 LOF missense mutations-induced effects on its autophosphorylation ability. The MST/MAP4K-8KO HEK293A cells were transfected with Myc-MAP4K3 and the indicated SFB-MAP4K3 LOF mutants and subjected to pulldown assay. A representative blot of two independent experiments is shown. **p, q** Characterization of the MAP4K2/3 LOF missense mutations-induced effects on their ATP-binding abilities. The MST/MAP4K-8KO HEK293A cells were transfected with the indicated SFB-tagged LOF mutants of MAP4K2 (**p**) and MAP4K3 (**q**), purified using S protein beads, washed with high-salt buffer containing 250 mM NaCl, and subjected to γ-ATP-based in vitro kinase assay. A representative blot of two independent experiments is shown. Source data are provided as a Source Data file.

---

for MAP4Ks activation and its phosphorylation status could be taken as a readout for MAP4K2/3 activities.

Indeed, all the MAP4K2 LOF mutations disrupted its autophosphorylation at S170 (Fig. 5k). As for the MAP4K3 LOF mutations, 3 of them (i.e., E61K, Y191C, R279W) significantly reduced MAP4K3 autophosphorylation at S170 while the rest 12 mutations fully abolished it (Fig. 5l). Since the E61K, Y191C, and R279W mutations completely abolished the ability of MAP4K3 to phosphorylate LATS1-HM like other MAP4K3 LOF mutations (Fig. 5e, g), we speculate that these three MAP4K3 mutants may lose the ability to recognize LATS substrate though they still have partial kinase activity (Fig. 5l), or they could be phosphorylated at S170 by other kinases due to the unique conformational changes caused by the mutations.

Next, we examined whether MAP4K2/3 LOF mutations could interfere with their autophosphorylation and ATP-binding abilities. As shown in Fig. 5m, all the MAP4K2 LOF mutations prevented its S170 phosphorylation as induced by wild-type MAP4K2 in the MST/MAP4K-8KO cells. Moreover, although co-expressing wild-type MAP4K3 significantly induced the S170 phosphorylation of MAP4K3 K45R mutant in the MST/MAP4K-8KO cells, it failed to further enhance the S170 phosphorylation for MAP4K3 E61K, Y191C and R279W LOF mutants (Fig. 5n). As for the rest 12 MAP4K3 LOF mutations (Fig. 5l), they all prevented MAP4K3 from being phosphorylated at S170 by wild-type MAP4K3 in the MST/MAP4K-8KO cells (Fig. 5o). Through the γ-ATP-based in vitro kinase assay, we showed that all the MAP4K2 LOF mutations abolished its ATP-binding ability (Fig. 5p). As for the MAP4K3 LOF mutations, 5 of them (i.e., E61K, R168W, E186Q, Y191C, R279W) did not affect MAP4K3 ATP-binding ability, while the other 10 LOF mutations dramatically inhibited it (Fig. 5q). These data together show that all the identified MAP4K2 LOF mutations can target both the autophosphorylation and ATP-binding abilities of MAP4K2, while the MAP4K3 LOF mutations can affect at least one of these two events for MAP4K3.

A similar MD analysis was performed for the MAP4K2/3 LOF mutations using the available protein structures of MAP4K2 (Supplementary Fig. 11a) and MAP4K3 (Supplementary Fig. 11g). The MAP4K2 LOF mutations all induced a conformational change for its kinase domain (Supplementary Fig. 11b), where its activation loop (Supplementary Fig. 11c) and helix region (Supplementary Fig. 11d) were significantly altered as compared to its β-sheet region (Supplementary Fig. 11e). The MAP4K2 LOF mutations also resulted in a fluctuation change across its kinase domain including the residues surrounding its autophosphorylation site S170 and ATP-binding site K45 (Supplementary Fig. 11f). As for MAP4K3, its LOF mutations caused a conformational change of its kinase domain (Supplementary Fig. 11h),

particularly for its activation loop (Supplementary Fig. 11i) as compared to its helix group (Supplementary Fig. 11j) and β-sheet region (Supplementary Fig. 11k). Moreover, 9 of the MAP4K3 LOF mutations significantly changed the fluctuation of the residues surrounding its autophosphorylation site S170 while this was not the case for the remaining 6 MAP4K3 LOF mutations (i.e., D196N, G201V, E206Q, T227P, L267V, P271L) (Supplementary Fig. 11l). We observed only a mild fluctuation change for the residues near the MAP4K3 ATP-binding site K45 of all the MAP4K3 LOF mutants (Supplementary Fig. 11l). Collectively, these MD data illustrate the potential effects on the MAP4K2/3 kinase domain structures as caused by their LOF mutations.

Our recent study revealed a Hippo-independent role of MAP4K2 in regulating autophagy and cell survival by binding and phosphorylating LC3[29]. Interestingly, one MAP4K2 neutral mutation, W356R, was found in its LC3-interacting region (LIR) motif (i.e., EEWTLL) (Supplementary Fig. 6d). Unlike other neutral mutations, W356R disrupted the interaction of MAP4K2 with LC3 in both wild-type and LATS1/2 DKO cells (Supplementary Fig. 6e). Moreover, MAP4K2 W356R mutant failed to localize onto autophagosome in chloroquine-treated wild-type and LATS1/2 DKO cells (Supplementary Fig. 6f). These findings suggest that the W356R neutral mutation inhibits the role of MAP4K2 in autophagy, highlighting potential YAP-independent roles of Hippo neutral mutations in cancer-related events.

## H161 is a crucial residue for MOB1 tumor suppressor function

Our Hippo missense mutation screen revealed H161Y and H161D as the sole LOF mutations for MOB1A and MOB1B, respectively (Fig. 2b and Supplementary Data 3). Indeed, in contrast to wild-type MOB1A/B, MOB1A-H161Y and MOB1B-H161D mutants failed to rescue YAP cytoplasmic localization (Fig. 6a), rescue YAP S127 phosphorylation (Fig. 6b), and suppress the transcription of YAP downstream genes *CTGF*, *CYR61* and *ANKRD1* (Fig. 6c) in the MOB1A/B DKO cells. These data confirm the negative effects of the H161Y and H161D mutations on MOB1A/B.

The MOB1A/B LOF mutations were both identified in head and neck cancer (HNC) patients (Fig. 2g), so we tested their oncogenic roles in HNC. To do so, we depleted MOB1A/B in a tongue cancer-derived cell line CAL-27 and reconstituted the MOB1A/B DKO CAL-27 cells with wild-type MOB1A and MOB1B and their H161Y and H161D mutants (Fig. 6d). Consistently, expressing wild-type MOB1A and MOB1B, but not their H161Y and H161D mutants, rescued YAP S127 phosphorylation in the MOB1A/B DKO CAL-27 cells (Fig. 6d). We then subjected these CAL-27 stable cells to an orthotopic xenograft assay and found that MOB1A/B deficiency promoted the CAL-27 xenograft

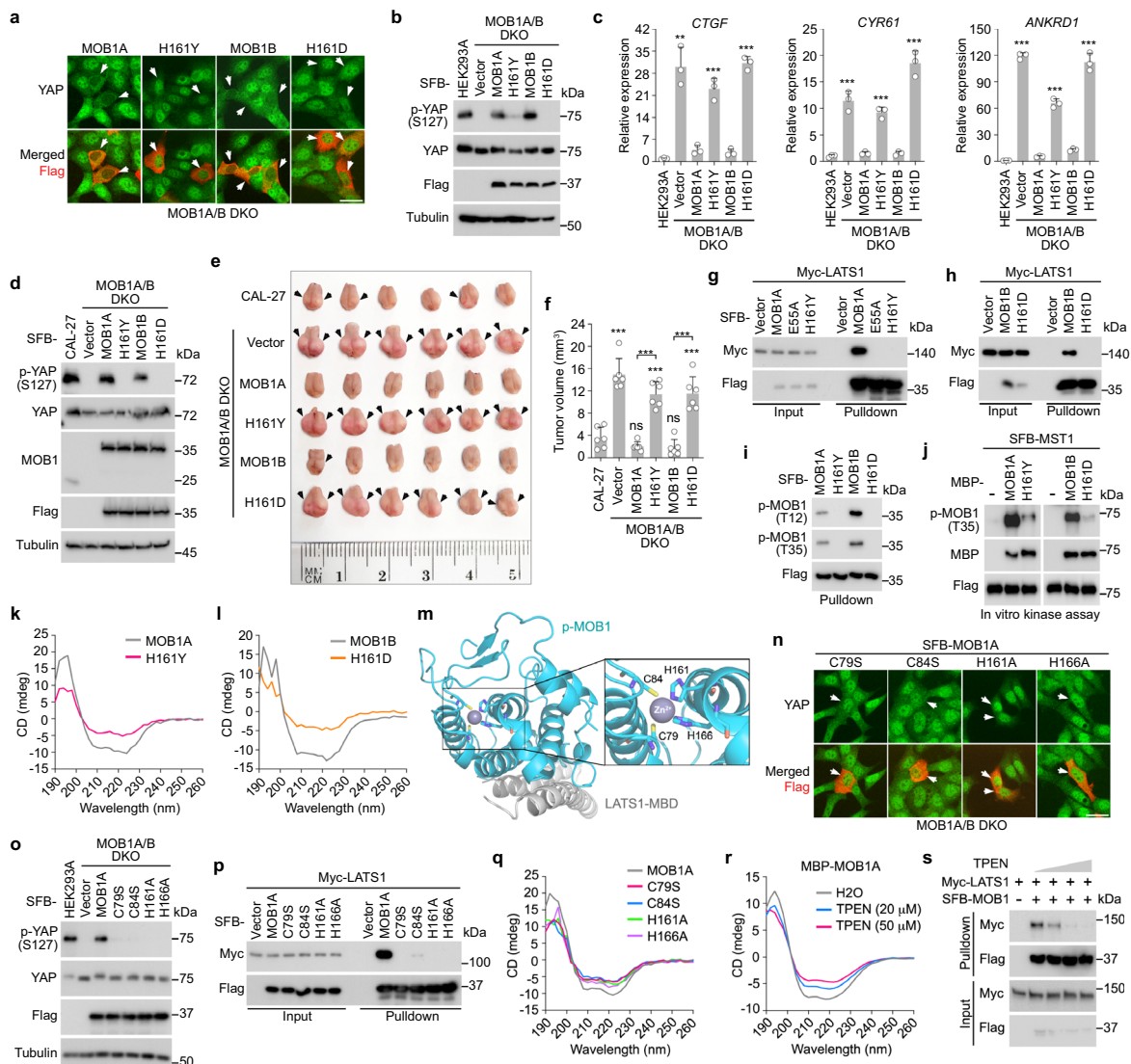

**Fig. 6 | Characterization of MOB1A/B LOF mutations reveals an essential role of the zinc finger (ZNF) domain for MOB1 function. a** The MOB1A/B LOF missense mutants fail to rescue YAP cytoplasmic localization in the MOB1A/B DKO cells. Scale bar, 30 μm. Arrows showed the cells expressing the indicated constructs. **b, c** The MOB1A/B LOF mutations inhibit MOB1A/B functions. The MOB1A/B DKO HEK293A cells were transduced with the indicated constructs and subjected to Western blot (**b**). A representative blot of two independent experiments is shown. The transcription of YAP downstream genes *CTGF*, *CYR61* and *ANKRD1* was examined by q-PCR (mean ± s.d., *n* = 3 biological replicates) (**c**). **\*\****p* < 0.01, **\*\*\****p* < 0.001 (two-tailed Student's *t*-test). **d** The MOB1A/B LOF mutants fail to rescue YAP S127 phosphorylation in the MOB1A/B DKO CAL-27 cells. A representative blot of two independent experiments is shown. **e, f** The MOB1A/B LOF mutants fail to inhibit MOB1A/B-deficient CAL-27 xenograft tumor growth in mouse tongues. The collected mouse tongues were shown (**e**). Tumor volume was measured and quantified (mean ± s.d., *n* = 6 mice per group) (**f**). ns, no significance. **\*\*\****p* < 0.001 (two-tailed Student's *t*-test). **g, h** The MOB1A/B LOF missense mutations inhibit their interaction with LATS1. A representative blot of two independent experiments is shown. **i** The MOB1A/B LOF missense mutations inhibit their phosphorylation at T12 and T35. A representative blot of two independent experiments is shown. **j** The MOB1A/

B LOF missense mutations inhibit MST1-mediated their T35 phosphorylation in vitro. A representative blot of two independent experiments is shown. **k, l** The MOB1A/B protein structures are changed by their LOF mutations. **m** Illustration of the MOB1 ZNF domain in the p-MOB1/LATS1-MBD co-crystal structure (PDB: 5BRK). **n–p** The MOB1A ZNF domain mutations target MOB1A function in the Hippo pathway. The MOB1A/B DKO HEK293A cells were transfected with the indicated constructs and subjected to immunofluorescent staining (**n**) and Western blot (**o**). Scale bar, 30 μm. Arrows showed the cells expressing the indicated constructs. HEK293T cells were transfected with Myc-LATS1 and the indicated SFB-MOB1A/B constructs and subjected to pulldown assay (**p**). A representative blot of two independent experiments is shown. **q** The MOB1A protein structure is changed by its ZNF domain site mutations. **r** MOB1A protein structure is changed by TPEN. MBP-tagged MOB1A protein was purified from bacteria, incubated with TPEN at the indicated concentrations, and subjected to circular dichroism analysis. **s** TPEN treatment disrupts the MOB1-LATS1 complex formation. Cell lysates were incubated with TPEN at different concentrations (i.e., 0 mM, 1 mM, 2 mM, and 4 mM) and subjected to pulldown assay. A representative blot of two independent experiments is shown. Source data are provided as a Source Data file.

tumor growth in mouse tongue tissues while reconstituting wild-type MOB1A and MOB1B, but not their H161Y and H161D mutants, inhibited tumor growth (Supplementary Fig. 12a and Fig. 6e, f).

Since the H161Y and H161D LOF mutations both occur at the H161 site, these findings indicate that H161 is a crucial site for MOB1 tumor suppressor function in the Hippo pathway.

## The zinc finger (ZNF) domain is an integral region of MOB1

To elucidate the mechanism, we compared the interacting proteins for wild-type MOB1A and MOB1B and their H161Y and H161D mutants via tandem affinity purification coupled with mass spectrometry (TAP-MS) analysis. Strikingly, we hardly identified any known MOB1-binding proteins, such as LATS1, LATS2, STK38, and STK38L, in the MOB1A-

H161Y and MOB1B-H161D-associated protein complexes (Supplementary Fig. 12b). These findings were further confirmed through a pull-down assay, where the H161Y and H161D mutations dramatically reduced the interaction of MOB1A/B with LATS1 (Fig. 6g, h), STK38 (Supplementary Fig. 12c, d), and STK38L (Supplementary Fig. 12e, f). Moreover, the H161Y and H161D mutations disrupted the MOB1A/B T12 and T35 phosphorylation in vivo (Fig. 6i) and MST1-induced MOB1A/B T35 phosphorylation in vitro (Fig. 6j). These data show that the H161Y and H161D mutations affect almost all the known functional events for MOB1.

We hypothesized that the H161 site could be involved in MOB1 protein folding, thus subjected the bacterially purified MOB1A-H161Y and MOB1B-H161B mutant proteins (Supplementary Fig. 12g) to CD analysis. Indeed, the H161Y and H161D mutations largely induced a conformational change of MOB1A (Fig. 6k) and MOB1B (Fig. 6l), respectively. Based on these findings, we turned to the p-MOB1/LATS1-MBD co-crystal structure[22] and surprisingly found that the H161 site is part of MOB1 zinc-finger (ZNF) domain (Fig. 6m). Interestingly, mutating either of the four ZNF domain sites C79, C84, H161 and H166 (i.e., C79S, C84S, H166A, H166A) (Fig. 6m) largely inhibited the ability of MOB1A to rescue YAP cytoplasmic localization (Fig. 6n) and S127 phosphorylation (Fig. 6o) in the MOB1A/B DKO cells. These ZNF domain mutations also disrupted the interaction of MOB1A with LATS1 (Fig. 6p) and changed its protein structure (Fig. 6q and Supplementary Fig. 12h). Given the essential role of zinc ion (Zn) in maintaining the ZNF domain, we further examined the role of Zn in regulating MOB1. Indeed, treatment with Zn chelator TPEN induced a conformational change of MOB1A (Fig. 6r) and reduced its interaction with LATS1 (Fig. 6s) in a dose-dependent manner. In addition, supplementing cadmium (Cd), a type of heavy metal known capable of substituting Zn from its cellular protein storage[30,31], induced a conformational change of MOB1A (Supplementary Fig. 12i) and disrupted the MOB1-LATS1 complex formation (Supplementary Fig. 12j). These data together demonstrate that the ZNF domain is an integral part of MOB1 required for its folding and function.

To gain a structural insight into the MOB1 LOF and ZNF domain mutations, we conducted a MD analysis using the p-MOB1/LATS1-MBD (Supplementary Fig. 13a) and MOB1/MST2-peptide (Supplementary Fig. 13e) co-crystal structures. Consistent with the CD data (Fig. 6k, l, q), the MOB1 LOF and ZNF domain mutations all changed the p-MOB1/LATS1-MBD complex structure (Supplementary Fig. 13b). Our RMSD data further revealed that the MOB1 LOF mutations H161Y and H161D resulted in a significant change in the MOB1 overall structure and its Zn interface but exerted only a mild effect on its helix group (Supplementary Fig. 13c). The conformation of the LATS1-MBD region was also slightly affected by the MOB1 LOF mutations (Supplementary Fig. 13c). We observed similar effects on MOB1, MOB1 Zn interface and LATS1-MBD of the p-MOB1/LATS1-MBD co-crystal structure as caused by the four MOB1 ZNF domain mutations (Supplementary Fig. 13d). These findings were confirmed using the MOB1/MST2-peptide co-crystal structure, whose conformation was changed by MOB1 LOF and ZNF domain mutations (Supplementary Fig. 13f), particularly for MOB1, its Zn interface and the MOB1-associated MST2 peptide (Supplementary Fig. 13g, h). Collectively, these data show an essential role of the ZNF domain in maintaining MOB1 structure.

## The NF2 LOF mutations induce an oncogenic role for NF2

Four NF2 LOF mutations (i.e., L46R, L141R, L208P, and A211D) (Fig. 2b and Supplementary Data 3) were identified in its FERM domain (Fig. 7a). Among them, the L46R, L141R and A211D mutations are known to inhibit NF2[32], while the L208P mutation has not been previously characterized. Consistently, all these NF2 LOF mutants failed to rescue YAP cytoplasmic localization (Fig. 7b) and S127 phosphorylation (Fig. 7c) in the NF2 KO cells. In contrast to previous findings[33–35],

the control NF2 S518A mutation did not affect the ability of NF2 to rescue YAP cytoplasmic localization (Fig. 7b) and S127 phosphorylation (Fig. 7c) in the NF2 KO cells. These findings suggest a dispensable role of S518 phosphorylation for NF2, consistent with a recent study[36]. Although phospholipids are known to bind and regulate NF2 through its FERM domain[23,36–38], we did not observe any changes in NF2 lipid-binding ability by its LOF mutations (Supplementary Fig. 14a–c). These LOF mutations largely reduced the interaction of NF2 with LATS1 (Fig. 7d) and AMOT (Fig. 7e), providing potential mechanisms for their inhibitory effects on NF2 in the Hippo pathway.

To determine whether the NF2 LOF mutants could dominant-negatively inhibit the Hippo pathway, we transduced NF2 and its LOF mutants into a non-transformed mammary epithelial cell line MCF10A (Fig. 7f) and subjected these stable cells to three-dimensional (3D) culture in Matrigel gel. Interestingly, expressing NF2 LOF mutants enhanced the MCF10A acini growth as compared to vector control and wild-type NF2 (Fig. 7g, h), while YAP S127 phosphorylation was not affected in their stable cells (Fig. 7f). These data suggest that the NF2 LOF mutations not only inhibit NF2 tumor suppressor role in the Hippo pathway, but also induce a YAP-independent oncogenic function for NF2.

## The NF2 LOF mutants promote tumorigenesis via the VANGL-JNK pathway

Next, we examined several cancer-related signaling pathways in the NF2 LOF mutants-transduced MCF10A cells and found that expressing the NF2 LOF mutants did not affect the activation of YAP, AKT or ERK but largely enhanced the phosphorylation of JNK and its substrate c-Jun (Fig. 7f). Moreover, we analyzed the NF2-associated high confident interacting proteins (HCIPs) based on two biologically repeated TAP-MS studies (Supplementary Fig. 14d). As shown in Fig. 7i and Supplementary Fig. 14e–s, the NF2 LOF mutations disrupted its interaction with many identified HCIPs including DCAF1, AP3B1, SF3BP2, LATS1, LUC7L, DDA1, FNBP4, CHD1, NOLC1 and ZC3H18; as for other HCIPs like DDX41, EPB41L5, PCCB, AP3M1, LUC7L3, RNPS1, their association with NF2 was either slightly changed or not consistently affected by its LOF mutations. Interestingly, VANGL1 is the only HCIP whose interaction with NF2 was largely induced by its LOF mutations (Fig. 7i, j). As planar cell polarity (PCP) pathway components, VANGL1 and its paralog protein VANGL2 are key regulators of epithelial polarity, development, and diseases[39,40]. Although VANGL2 also strongly interacted with the NF2 LOF mutants (Supplementary Fig. 14t), it was not identified by our TAP-MS analysis due to its low expression in HEK293A cells (Supplementary Fig. 14u). Low expression of VANGL2 was also found in MCF10A cells (Supplementary Fig. 14u).

VANGL1/2 and their associated PCP pathway are known to activate JNK in development and tumorigenesis[41,42], raising the possibility that the NF2 LOF mutants may activate JNK through VANGL1/2. Indeed, downregulation of VANGL1 largely reduced the phosphorylation of JNK and c-Jun in the MCF10A cells stably expressing the NF2 LOF mutants (Fig. 7k). Loss of VANGL1 or treatment with JNK inhibitor JNK-IN-8 (Fig. 7l) dramatically attenuated the overgrowth phenotype for the NF2 LOF mutants-transduced MCF10A acini (Fig. 7m, n). Since the NF2 LOF mutations were all identified in the meningioma patients (Fig. 7a), we further examined their tumorigenic roles using a malignant meningioma cell line IOMM-Lee. To do so, the NF2 KO IOMM-Lee cells were generated and reconstituted with the NF2 LOF mutants, where the expression level of the exogenously expressed NF2 LOF mutants was made close to that of NF2 in the wild-type cells (Fig. 7o). Although reconstituting the NF2 LOF mutants failed to rescue YAP S127 phosphorylation in the NF2 KO IOMM-Lee cells, it significantly induced the phosphorylation of JNK and c-Jun (Fig. 7o). In addition, the NF2 LOF mutants promoted the NF2 KO IOMM-Lee xenograft tumor growth, while treatment with JNK inhibitor JNK-IN-8 significantly attenuated it (Fig. 7p, q). Taken together, these results show that the NF2 LOF

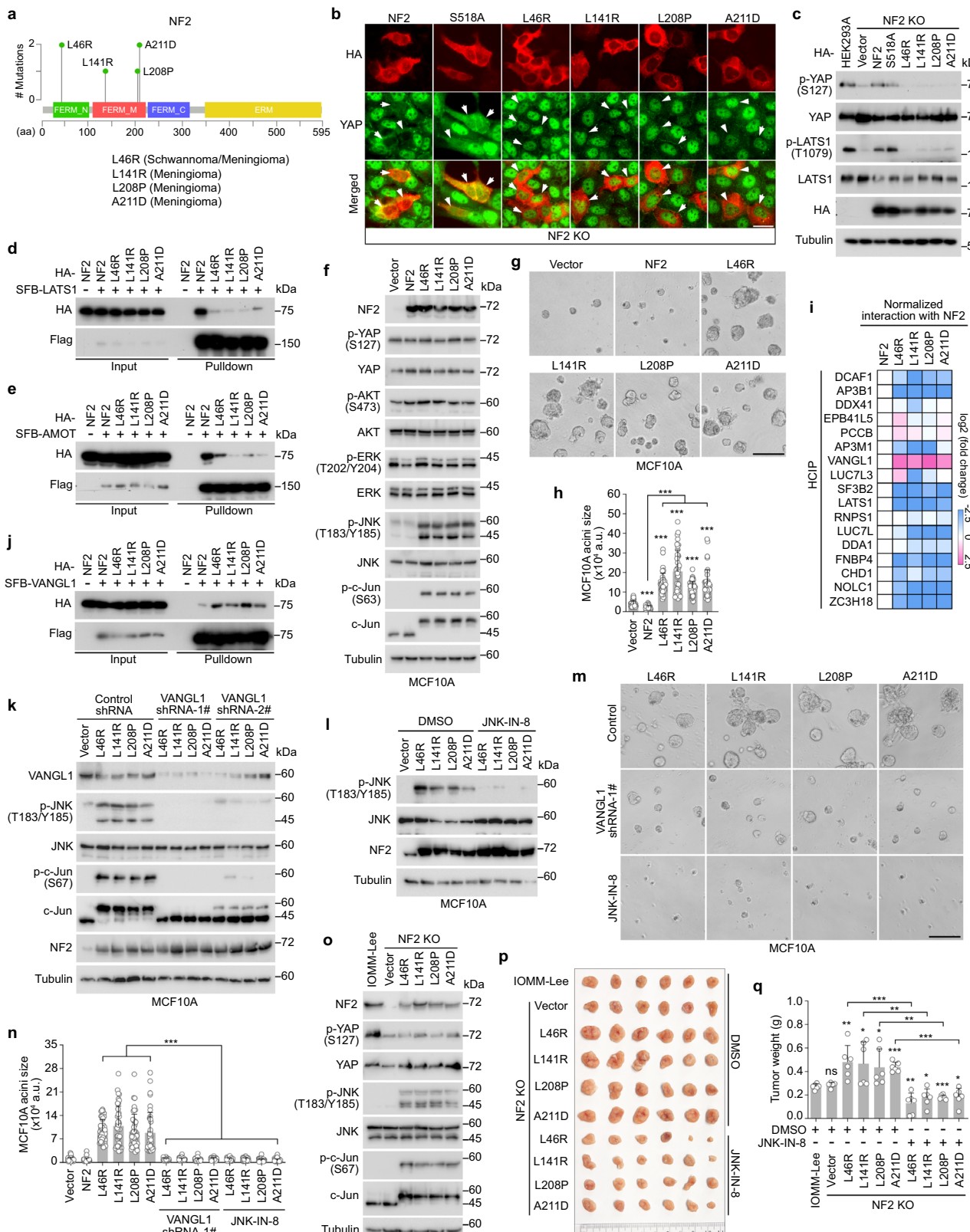

mutants induce cell transformation and tumor growth by activating the VANGL-JNK pathway.

## Discussion

Here, we focused on the human cancer genome-derived Hippo somatic mutations to elucidate how the Hippo pathway becomes dysregulated in human cancers. Although YAP/TAZ are known to

function as tumor suppressors in specific types of human cancers and the Hippo pathway can exert YAP/TAZ-independent functions in several cancer-related signaling events, the Hippo pathway LOF mutations identified here are defined based on its classic tumor suppressive role by inhibiting YAP and TAZ.

It is notable that a limited number of LOF missense mutations were identified for the major Hippo pathway genes. This may be due to

**Fig. 7 | The NF2 LOF mutations induce an oncogenic role of NF2 through the VANGL-JNK pathway. a** Illustration of NF2 domains and its identified LOF missense mutations. **b** The NF2 LOF mutants fail to rescue YAP cytoplasmic localization in the NF2 KO cells. Scale bar, 30 μm. Arrows showed the cells expressing the indicated constructs. **c** Reconstituting the NF2 LOF mutants fail to rescue YAP S127 phosphorylation in the NF2 KO cells. A representative blot of two independent experiments is shown. **d, e** The NF2 LOF mutations inhibit its interaction with LATS1 and AMOT. HEK293T cells were transfected with the indicated HA-NF2 constructs and SFB-tagged LATS1 (**d**) or AMOT (**e**) and subjected to pulldown assay. A representative blot of two independent experiments is shown. **f** NF2 LOF mutants induces the phosphorylation of JNK and c-Jun in MCF10A cells. A representative blot of two independent experiments is shown. **g, h** The NF2 LOF mutants promote MCF10A acini growth. The representative images were shown (**g**). The acini size was measured and quantified (mean ± s.d., *n* = 40 acini per group) (**h**). ***p < 0.001 (two-tailed Student's *t*-test). Scale bar, 400 μm. **i, j** The NF2 LOF mutations enhance its interaction with VANGL1. The interaction between the indicated HCIPs and NF2 LOF mutants was normalized and shown as a heatmap (**i**). HEK293T cells were transfected with SFB-VANGL1 and the indicated HA-NF2 LOF mutants and subjected to

pulldown assay (**j**). A representative blot of two independent experiments is shown. **k** The NF2 LOF mutants activate JNK through VANGL1. The MCF10A cells stably expressing NF2 and its LOF mutants were transduced with control and VANGL1 shRNAs. A representative blot of two independent experiments is shown. **l** The MCF10A cells stably expressing the NF2 LOF mutants were treated with JNK-IN-8 (10 μM) for 4 h and subjected to Western blot. A representative blot of two independent experiments is shown. **m, n** VANGL1 and JNK are required for the NF2 LOF mutants-induced MCF10A acini growth in Matrigel. The representative images were shown (**m**). The acini size was measured and quantified (mean ± s.d., *n* = 40 acini per group) (**n**). ***p < 0.001 (two-tailed Student's *t*-test). Scale bar, 400 μm. **o** The NF2 LOF mutants activate JNK in IOMM-Lee cells. A representative blot of two independent experiments is shown. **p, q** Treatment with JNK-IN-8 inhibits NF2 LOF mutants-induced IOMM-Lee xenograft tumor growth. The collected tumors treated with DMSO or JNK-IN-8 (10 mg/kg) were shown (**p**) and tumor weight was measured and quantified (mean ± s.d., *n* = 6 mice per group) (**q**). ns, no significance. *p < 0.05, **p < 0.01, ***p < 0.001 (two-tailed Student's *t*-test). Source data are provided as a Source Data file.

the stringent strategy used in our Hippo LOF mutation screen. We re-expressed each Hippo gene missense mutant in their corresponding KO cells and used clear YAP nuclear-to-cytoplasmic translocation as a readout to determine their functions in the Hippo pathway. This approach may result in missing the LOF mutations that partially affect the Hippo pathway genes' activities and those affecting protein stability. Moreover, the Hippo pathway can be affected by different alterations in cancer (e.g., gene fusion, frameshift, splicing) and is not limited to missense mutation (Fig. 1f). Dysregulation of Hippo pathway regulators, such as *GNAQ/GNA11*[43,44] and PTPN14[45], also leads to Hippo pathway inactivation in cancer. These facts highlight the complex reasons for Hippo pathway inactivation in human cancer development, making it different from other cancer-related genes like *TP53*, *PI3KCA*, *KRAS*, where missense mutations are the predominant driver alteration.

Although individual Hippo genes are infrequently altered in human cancers (Fig. 1f), the overall oncogenic alteration rate of Hippo signaling is comparable to that of several cancer-related genes (Supplementary Fig. 5a, b). Notably, this analysis did not include the LOF missense mutations for Hippo core components SAV1 and MAP4K1/4/6/7, as well as the LOF mutations for Hippo periphery regulators, such as *WWC1-3*, *FAT1-4*, *PTPN14*, *AMOT/L1/L2*. These facts underscore the broad impact of dysregulated Hippo signaling on human cancer development. Although significant progress has been made in developing compounds targeting YAP/TAZ and TEAD1-4, biomarkers that can be used to select patients for treatment with Hippo pathway drugs are limited. Since Hippo pathway deficiency can induce cancer addiction to YAP[43,46–48], the identified Hippo LOF mutations could be used as genomic biomarkers and developed as a "toolset" to predict YAP activation and guide the application of YAP/TAZ/TEAD-based compounds for personalized cancer therapy.

Our somatic mutation study also unveiled additional regulations and functions for the Hippo pathway components. We not only discovered an essential role of the ZNF domain for MOB1 protein folding and function (Fig. 6 and Supplementary Fig. 12, 13), but also highlighted the critical role of Zn in regulating MOB1 (Fig. 6r, s). Heavy metals (e.g., Cd, Zn) have been shown to inhibit the Hippo pathway[49], while the underlying mechanisms have not been fully understood. Here, our MOB1 LOF mutation study provides mechanistic insight into the knowledge gap. Moreover, we uncovered an oncogenic role of the NF2 LOF mutants mediated by PCP pathway component VANGL1 and its downstream JNK signaling. Interestingly, elevated JNK activity has been observed in NF2-mutated cancer like schwannoma[50], suggesting a pathological relevance of this finding. It is still unclear how the NF2 LOF mutations promote the interaction of NF2 with VANGL to activate JNK. Previous studies show that VANGL activates JNK through the Wnt

pathway component DVL[51] and autophagy protein p62/SQSTM1[41]. Whether the NF2 LOF mutants could activate the VANGL-JNK pathway through DVL and/or p62 deserves further investigation. Notably, the identified NF2 LOF mutations are all in NF2 FERM domain (Fig. 7a), implicating a role of the altered FERM domain in this process. Although not included in TCGA or COSMIC, additional NF2 FERM domain-associated mutations (e.g., F62S, L64P, G197C, E270G) also inhibited NF2[32,52]. It will be interesting to determine whether these NF2 mutations possess similar oncogenic function and how the altered NF2 FERM domain promotes its interaction with VANGL to activate JNK. Our study also revealed a total of 945 neutral mutations in the Hippo pathway components (Fig. 2a, c, e and Supplementary Data 3). One caveat of our screen strategy is its inability to uncover the neutral mutations that can further promote the Hippo pathway activation. Notably, the Hippo pathway plays oncogenic roles in several types of human cancers, such as estrogen receptor-positive breast cancer[53–55], hematologic malignancies (e.g., lymphomas)[56], prostate cancer[57], and small cell lung cancer[58]. Therefore, it will be highly interesting to analyze the Hippo neutral mutations in these cancer types and determine whether any of these mutations could activate the Hippo pathway to induce tumorigenesis. Completion of such a neutral mutation analysis would help better understand the Hippo pathway dysregulations in human cancer development.

## Methods
### Animal experiments
Athymic nude (nu/nu) mouse strain was used for the xenograft tumor assays in this study. All the nude mice were purchased from Jackson Laboratory (002019) and kept in a pathogen-free environment in ULAR Facility at University of California, Irvine under a 14-h light/10-h dark cycle with temperature of 75 °F and 60% humidity. All the tumor assays including the maximal tumor size/burden were followed with institutional guidelines, approved by the Institutional Animal Care and Use Committee (IACUC; protocol number AUP-19-112) of the University of California, Irvine, and performed under veterinary supervision. The maximal tumor size/burden was not exceeded during the animal experiments.

As for the MOB1 orthotopic xenograft tumor study, the indicated CAL-27 cells ($3 \times 10^5$) were injected into the tongue tissues of eight-week-old female nude mice. Mice were euthanized for tumor collection and analysis around 15 ~ 17 days post injection. The tumor volume was calculated using the standard formula V = length × width$^2$/2.

As for the NF2 xenograft tumor study, the indicated IOMM-Lee cells ($2 \times 10^6$) were subcutaneously injected into the four-week-old female nude mice. When tumors were approximately 50 mm$^3$ in size, 12 mice for each cell line were randomly assigned into two groups (6 mice

per group) and subjected to the indicated treatment (vehicle control or 10 mg/kg JNK-IN-8) via intraperitoneal administration every other day. After 15 days for adaptation, mice were euthanized, and tumor weights were analyzed.

## Cell lines

HEK293T (a female cell line, ATCC: CRL-3216) and MCF10A (a female cell line, ATCC: CRL-10317) were purchased from ATCC and kindly provided by Dr. Junjie Chen (MD Anderson Cancer Center). HEK293A (a female cell line, Thermo Fisher Scientific: R70507) was kindly provided by Dr. Jae-Il Park (MD Anderson Cancer Center). CAL-27 (a male cell line, ATCC: CRL-2095) and IOMM-Lee (a male cell line, ATCC: CRL-3370) were purchased from ATCC. HEK293T, HEK293A, CAL-27, and IOMM-Lee cells were maintained in Dulbecco's Modified Eagle's Medium (DMEM) supplemented with 10% fetal bovine serum at 37 °C in 5% $CO_2$ (v/v). MCF10A cells were maintained in DMEM/F12 medium supplemented with 5% horse serum, 200 ng/mL epidermal growth factor, 500 ng/mL hydrocortisone, 100 ng/mL cholera toxin and 10 μg/mL insulin at 37 °C in 5% $CO_2$ (v/v). All the culture media contain 1% penicillin and streptomycin.

## Antibodies and chemicals

For Western blot, anti-Flag (M2)-peroxidase (HRP) (A8592-1MG, 1:5000 dilution), anti-α-tubulin (T6199-200UL, 1:5000 dilution) and anti-β-actin (A5441-100UL, 1:5000 dilution) antibodies were obtained from Sigma-Aldrich. Anti-Myc (sc-40, 1:500 dilution) and anti-GFP (sc-390394, 1:1000 dilution) antibodies were purchased from Santa Cruz Biotechnology. Anti-hemagglutinin (HA) antibody (MMS-101P, 1:3000 dilution) was obtained from BioLegend. Anti-phospho-YAP (Ser127) (4911S, 1:1000 dilution), anti-phospho-LATS1 (Thr1079) (8654S, 1:1000 dilution), anti-phospho-MST1 (Thr183)/MST2 (Thr180) (3681S, 1:1000 dilution), anti-phospho-MOB1 (Thr12) (8843S, 1:1000 dilution), anti-phospho-MOB1 (Thr35) (8699S, 1:1000 dilution), anti-LATS1 (3477S, 1:2000 dilution), anti-NF2 (12896S, 1:2000 dilution), phospho-AKT (Ser473) (4060S, 1:1000 dilution), anti-AKT (4691S, 1:2000 dilution), anti-phospho-p44/42 MAPK (Erk1/2) (Thr202/Tyr204) (4370S, 1:1000 dilution), anti-p44/42 MAPK (Erk1/2) (9102S, 1:2000 dilution), anti-phospho-JNK (Thr183/Tyr185) (9251S, 1:1000 dilution), anti-JNK (9252S, 1:2000 dilution), anti-phospho-c-Jun (Ser63) (9261S, 1:1000 dilution), anti-c-Jun (9165 T, 1:2000 dilution), and anti-VANGL1 (14783S, 1:1000 dilution) antibodies were purchased from Cell Signaling Technology. ATP-γ-S kinase substrate (ab138911), p-Nitrobenzyl mesylate (ab138910), and anti-Thiophosphate ester antibody (ab92570, 1:1000 dilution) were obtained from Abcam. Anti-phospho-MAP4K2 (Ser170) (AB-PK646, 1:1000 dilution) was purchased from Kinexus. Anti-phospho-MAP4K3 (Ser170) (1:500 dilution) was raised against keyhole limpet hemocyanin (KLH)-conjugated phospho-peptide CATIAKRK(pSer)FIGTPYW and affinity purified using Sulfo-Link peptide Coupling Resin (Thermo Fisher Scientific). Anti-GST (1:5000 dilution) and anti-MBP (1:5000 dilution) antibodies were raised by immunizing a rabbit with bacterially expressed and purified GST and MBP full-length proteins, respectively. Anti-YAP antibody (1:1000 dilution) was raised by immunizing a rabbit with bacterially expressed and purified GST-YAP full-length fusion protein as described previously[59]. All the antisera were affinity-purified using AminoLink Plus Immobilization and Purification Kit (Pierce).

For immunofluorescent staining, anti-YAP antibody (sc-101199, 1:200 dilution) was purchased from Santa Cruz Biotechnology. Anti-Flag (F7425-.2MG, 1:5000 dilution) antibody was purchased from Sigma-Aldrich. Anti-HA (3724S, 1:500 dilution) antibody was purchased from Cell Signaling Technology.

For chemicals, CaCl2 (C7902), CdCl2 (655198), TPEN (P4413) were purchased from Sigma-Aldrich. JNK-IN-8 (HY-13319) was obtained from MedChemExpress. Chloroquine (193919) was obtained from MP Biomedicals.

## Constructs and viruses

Plasmids encoding the indicated genes were obtained from the Human ORFeome V5.1 library or purchased from the DNASU Plasmid Repository. All constructs were generated via polymerase chain reaction (PCR) and sub-cloned into a pDONOR201 vector using Gateway Technology (Thermo Fisher Scientific) as entry plasmids. Gateway-compatible destination vectors with the indicated SFB tag, Myc tag, HA tag, GST tag, MBP tag and GFP tag were used to express various fusion proteins. For tandem affinity purification (TAP), entry plasmids were subsequently recombined into a lentiviral gateway-compatible destination vector for the expression of SFB-tagged fusion proteins. PCR-based mutagenesis was used to generate the indicated site mutations.

The VANGL1 pLKO.1 shRNAs were purchased from Sigma-Aldrich. The sequence information of shRNAs used for VANGL1 knockdown studies is as follows:

VANGL1 shRNA-1# (TRCN0000062092): CTCGTAGTCAATGTGAA GAAA;

VANGL1 shRNA-2# (TRCN0000290422): CCATTCATCATACTCTC TGAA;

Control shRNA (Addgene plasmid # 136035): CCTAAGGTTAAGT CGCCCTCG.

Lentiviral supernatants were generated by transient transfection of HEK293T cells with the helper plasmids pSPAX2 and pMD2G and harvested 48 h later. Supernatants were passed through a 0.45-μm filter and used to infect cells with the addition of 8 μg/mL hexadimethrine bromide (Polybrene) (Sigma-Aldrich). Plasmid transfection was performed using polyethyleneimine (PEI) (23966-2, Polysciences).

## Bioinformatic analysis of the Hippo pathway somatic mutations

The Hippo pathway gene alteration data were obtained from the cBioportal database (http://www.cbioportal.org), where 89 TCGA studies comprising 32 types of human cancers and 11706 unique patient samples (http://bit.ly/2DTHUfr) were selected for the bioinformatic analysis. All the Hippo pathway gene alteration data and their associated patient samples were listed in Supplementary Data 1. All the Hippo pathway gene somatic mutations were listed in Supplementary Data 2. All the Hippo pathway gene missense mutations including the ones of NF2 in meningioma and schwannoma obtained from COSMIC were listed in Supplementary Data 3. The identified Hippo LOF missense mutations in this study were also listed in Supplementary Data 3. The oncogenic alteration information for a group of oncogenes and tumor suppressor genes was listed in Supplementary Data 4. Specifically, for *TP53*, *PI3KCA*, *PTEN*, *KRAS* and *HRAS*, their driver alteration information was obtained from cBioportal; for YAP, TAZ and TEAD1-4, their oncogenic alterations included the ones documented in cBioportal and the reported missense mutations obtained from a previous study[14]; for the Hippo pathway genes, their oncogenic alterations included the LOF missense mutations revealed in this study and the ones obtained from cBioportal including the homogeneous deletion (HOMDEL), nonsense mutations, fusion mutations, frame-shift/insertion/deletion mutations, splicing-region/site mutations, translation start site mutations.

## Gene inactivation by the CRISPR/Cas9 system

To generate knockout cells, five distinct single-guide RNAs (sgRNA) were designed by the CHOPCHOP website (https://chopchop.rc.fas.harvard.edu), cloned into lentiGuide-Puro vector (Addgene plasmid # 52963), and transfected into cells with lentiCas9-Blast construct (Addgene plasmid # 52962). The next day, cells were transiently selected with puromycin (2 μg/ml) for two days and sub-cloned to form single colonies. Knockout cell clones were screened by Western blot to verify the loss of target protein expression. The LATS1/2 DKO, MOB1A/B DKO, MST1/2 DKO, and NF2 KO HEK293A cells were generated as described previously[23], and the related sgRNA sequence

information was included in Supplementary Data 5. The MST/MAP4K-8KO HEK293A cells were kindly provided by Drs. Zhipeng Meng (University of Miami) and Kun-Liang Guan (Westlake University). The CAL-27 MOB1A/B DKO cells and the IOMM-Lee NF2 KO cells were generated using the same sgRNAs as listed in Supplementary Data 5.

## Molecular dynamics simulations

Crystal structures of the human MST1/2 kinase domains, p-MOB1/LATS1-MBD, MOB1-MST2 peptide, and MAP4K3 kinase domain were retrieved from the Protein Data Bank (PDB) with respective IDs: 3COM, 4LG4[60], 5BRK[22], 5BRM[22], and 5J5T[61]. Any absent loop residues within these structures were modeled and completed using MODELLER[62]. In instances where PDB structures for the human LATS1/2 and MAP4K2 kinase domains were unavailable, predictions were carried out using the AlphaFold2 algorithm[63]. Specific residues were subjected to mutagenesis using PyMOL to derive mutant constructs. N- and C-termini of all protein systems were capped using ACE and NHE groups, respectively.

All simulations were performed using the PMEMD program implemented in the AMBER22 molecular dynamics suite (https://ambermd.org/). Parameterization was conducted with the LEaP module in AMBER22, employing the protein force field ff14SB[64]. For the zinc binding interface (ZN-CCHH), the nonstandard force field parameters encompassing atom types like ZN (sp3), S4 (sp3), and N3 (sp3) for the ZAFF metal center were used[65]. The atomic ion library was incorporated along with the parameterization files for monovalent metal ions. For residues like phosphorylated threonine (TPO), additional parameters from GAFF[66] and PHOSAA10[67] were utilized. The protein was capped with ACE and NHE groups at the N- and C-termini, respectively. All systems were solvated in a TIP3P truncated octahedron water box, and neutralized with either Na+ or Cl− counter ions and with additional NaCl added to 0.15 M. These systems were first minimized, followed by a heating process to 310 K for 100 ps, with a time step of 2 fs in the canonical (NVT) ensemble. They were then equilibrated for 10 ns at 310 K with time step of 2 fs in the isothermal−isobaric (NPT) ensemble. The final production runs were performed in the NVT ensemble with a time step of 2 fs at 310 K for 500 ns. Three independent trajectories with identical starting structures and random initial velocities were generated for the wild-type, mutant, and phosphorylated complexes to enhance sufficient sampling. For all simulations, the last 200 ns trajectory was used for the subsequent analysis to ensure converged calculations. The Root Mean Square Deviation (RMSD) and Root Mean Square Fluctuation (RMSF) values were calculated using CPPTRAJ[68] against the averaged wild-type structure with Cα atoms. Snapshots closest to the average structures were used as the representation of the molecular dynamics-generated conformations. Structural visualization and alignment were facilitated by PyMOL.

## Immunofluorescent staining

Immunofluorescent staining was performed as described previously[69]. Briefly, cells cultured on coverslips were fixed with 4% paraformaldehyde for 10 min at room temperature and then extracted with 0.5% Triton X-100 solution for 5 min. After blocking with Tris-buffered saline with Tween 20 (TBST) containing 1% bovine serum albumin, cells were incubated with the indicated primary antibodies for 1 h at room temperature. After that, cells were washed and incubated with fluorescein isothiocyanate- or rhodamine-conjugated secondary antibodies for 1 h. Cells were counterstained with 100 ng/mL 4′,6-diamidino-2-phenylindole (Dapi) for 2 min to visualize nuclear DNA. The cover slips were mounted onto glass slides with anti-fade solution and visualized under a Nikon Ti2-E inverted microscope.

For YAP-based immunofluorescence screen study, Hippo gene missense mutants were transiently transfected into their corresponding Hippo KO cells (~30% confluence) overnight, serum starved for 5−6 h, and subjected to immunofluorescent staining using anti-YAP and the indicated tag antibodies.

## In vitro kinase assay

SFB-tagged Hippo pathway kinases (i.e., LATS1, LATS2, MST1, MST2, MAP4K2, MAP4K3) and their indicated mutants were expressed in HEK293T cells and purified using S protein beads. The isolated kinases were washed three times with the washing buffer (40 mM HEPES, 250 mM NaCl) and once with the kinase buffer (30 mM HEPES, 50 mM potassium acetate, 5 mM MgCl2), and subjected to the kinase assay in the presence of cold ATP (500 μM). Bacterially purified GST-tagged YAP or MBP-tagged LATS1-C3 region[23] was used as substrate. The reaction mixture was incubated at 30 °C for 30 min, terminated with 5× SDS loading buffer, and subjected to sodium dodecyl sulfate polyacrylamide gel electrophoresis (SDS-PAGE). The LATS1/2-induced GST-YAP phosphorylation was examined using anti-phospho-YAP (S127) antibody. Phosphorylation of MBP-LATS1-C3 was determined by anti-phospho-LATS1 (T1079) antibody.

ATP-γ-S-based kinase assay was modified to detect kinase ATP binding ability. SFB-tagged LATS1/2, MST1/2, MAP4K2/3 and their indicated missense mutants were expressed in the indicated cells for 48 h, purified using S protein beads, washed three times with washing buffer (40 mM HEPES, 250 mM NaCl) and once with the kinase buffer (30 mM HEPES, 50 mM potassium acetate, 5 mM MgCl2), and subjected to kinase assay in the presence of 500 μM ATP-γ-S. The reaction mixture was incubated at 30 °C for 30 min, supplemented with 2.5 mM p-Nitrobenzyl mesylate (PNBM) at room temperature for 1 h, terminated with 5× SDS loading buffer, and subjected to SDS−PAGE. ATP binding was detected by anti-Thiophosphate-ester antibody.

## RNA extraction, reverse transcription, and real-time PCR

RNA samples were extracted with TRIzol reagent (Invitrogen). Reverse transcription assay was performed with the Script Reverse Transcription Supermix Kit (Bio-Rad) according to the manufacturer's instructions. Real-time PCR was performed using Power SYBR Green PCR master mix (Applied Biosystems). For quantification of gene expression, the $2^{-\Delta\Delta Ct}$ method was used. *GAPDH* expression was used for normalization. The sequence information of q-PCR primers used for gene expression analysis was listed in Supplementary Data 5.

## Circular dichroism (CD) measurement

CD spectroscopy in the UV region (190 nm ~ 260 nm) was used to analyze protein secondary structure. Briefly, 0.2 μg/μL MBP-MST1/2 kinase domain proteins were dialyzed and prepared in the CD buffer (137 mM NaF, 2.7 mM KCl, 10 mM phosphate, pH 7.4) in a 1-mm cuvette (NSG Precision Cells, Inc., Farmingdale, NY). TPEN or CaCl2 at the indicated concentrations were added to the CD buffer for MBP-MOB1A CD measurement. The CD data were collected using a Jasco J-810 spectropolarimeter (JASCO, Easton, MD) with a scan speed of 50 nm/min and 2-nm wavelength intervals at 25 °C.

## Tandem affinity purification coupled with mass spectrometry (TAP-MS) analysis

The TAP-MS analysis was performed as described previously[70,71]. Briefly, HEK293A cells stably expressing SFB-tagged MOB1A, MOB1A-H161Y, MOB1B, MOB1-H161D, and NF2 were generated by culturing in medium containing 2 μg/mL puromycin and validated by immunostaining and Western blot. The stable cells were lysed in NETN buffer containing protease and phosphatase inhibitors at 4 °C for 20 min. The crude lysates were centrifuged at 15,700 g at 4 °C for 15 min. The supernatants were incubated with streptavidin-conjugated beads (GE Healthcare) at 4 °C for 4 ~ 6 h. The beads were then washed three times with NETN buffer, and the bound proteins were eluted with NETN buffer containing 2 mg/mL biotin (Sigma-Aldrich) at 4 °C overnight. The elutes were incubated with S protein beads (Novagen) at 4 °C for

4 h. The beads were washed three times with NETN buffer and subjected to SDS-PAGE. Each sample was run into the separation gel for a short distance, so that the whole bands could be excised as one sample.

The excised gel sample was cut into approximately 1-mm³ pieces. The gel pieces were then subjected to in-gel trypsin digestion[72] and dried. Samples were reconstituted in 5 μL of high-performance liquid chromatography (HPLC) solvent A (2.5% acetonitrile, 0.1% formic acid). A nanoscale reverse-phase HPLC capillary column was created by packing 5-μm C18 spherical silica beads into a fused silica capillary (100 μm inner diameter × ~20 cm length) with a flame-drawn tip. After the column was equilibrated, each sample was loaded onto the column via a Famos autosampler (LC Packings). A gradient was formed, and peptides were eluted with increasing concentrations of solvent B (97.5% acetonitrile, 0.1% formic acid). As the peptides eluted, they were subjected to electrospray ionization and then entered into an LTQ Orbitrap Elite mass spectrometer (Thermo Fisher Scientific). The peptides were detected, isolated, and fragmented to produce a tandem mass spectrum of specific fragment ions for each peptide. Peptide sequences (and hence protein identity) were determined by matching protein databases with the fragmentation pattern acquired by the software program SEQUEST (ver. 28) (Thermo Fisher Scientific). Enzyme specificity was set to partially tryptic with 2 missed cleavages. Modifications included carboxyamidomethyl (cysteines, fixed) and oxidation (methionine, variable). Mass tolerance was set to 5 ppm for precursor ions and 0.5 Da for fragment ions. The database searched was UniProt. Spectral matches were filtered to contain a false discovery rate of less than 1% at the peptide level using the target-decoy method[73], a simple and powerful way to deliver false positive estimations applied to MS/MS workflow. The protein inference was considered followed the general rules[74] with manual annotation based on experience applied when necessary. Briefly, computational tools, protein sequence databases, identification of mature forms of proteins, quantitative proteomics, integration of proteomic and transcriptional data, integration of multiple shotgun proteomic datasets, and gene-centered data interpretation were considered for protein inference. This same principle was used for isoforms when they were present in the database. The longest isoform was reported as the match.

### PIP strip binding assay
The PIP Strips (P-6001) were purchased from Echelon Biosciences. The binding assay was performed according to the manufactory's manual with slight modifications. Briefly, the PIP Strip membranes were blocked in TBST buffer containing 3% fatty acid-free bovine serum albumin at 4 °C overnight and incubated with 0.5 μg/mL bacterially purified MBP-NF2 FERM domain proteins at room temperature for 2 h. After it, the membranes were washed three times with TBST buffer and subjected to primary antibody incubation. Western blot was performed using anti-MBP antibody to detect the lipid-protein interaction.

### MCF10A acini three-dimensional (3D) growth assay
The MCF10A acini 3D growth assay was performed as previously described[75]. Briefly, $5 \times 10^3$ MCF10A cells were grown in 0.5% growth factor-reduced Matrigel (BD Biosciences) and seeded on the Matrigel pretreated eight-well chamber slide system (Fisher Scientific) for 4 days. The acini were imaged, and their relative size was calculated using Image J software.

### Statistics and reproducibility
Each experiment was repeated twice or more, unless otherwise noted. There were no samples or animals excluded from the analyses in this study. There was no statistical method used to predetermine the sample size for the mouse experiments. We assigned the animals randomly to different groups. A laboratory technician was blinded to the group allocation and tissue collections during the animal experiments as well as the data analyses. The Student's $t$-test (two-tailed) was used to analyze the differences between two sample groups. The chi-squared and log-rank (Mantel-Cox) tests were used to analyze the clinical data of the patients with normal and altered Hippo signaling genes. SD was used for error estimation. A $p$ value < 0.05 was considered statistically significant.

### Reporting summary
Further information on research design is available in the Nature Portfolio Reporting Summary linked to this article.

## Data availability
The proteomic data generated in this study have been deposited in the ProteomeXchange Consortium database via the PRIDE partner repository with the dataset identifier PXD049472: Project Name: Human Hippo cancer mutation proteins TAP-LC-MS; Project accession: PXD049472; Project https://doi.org/10.6019/PXD049472. Source data are provided with this paper.

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

## Acknowledgements

We thank Dr. Ross Tomaino (Taplin Mass Spectrometry Facility, Harvard Medical School) for the mass spectrometry analysis and the UCI Bio Sci 199 program undergraduate students (Jiadong Yang, Yifan Zhao, Kathern Le Nguyen, Amell Taffy Bishara, Tejas Krishen Mandalia, Kimberly Chuc, Yongqi Lin, Kevin Nguyen) for the help with Hippo missense mutant cloning. This work was supported by National Natural Science Foundation of China grant (32370766) and National Natural Science Fund for Excellent Young Scientists Fund Program (Overseas) to H.H., NIH grants (R01GM126048, R01GM143233) and American Cancer Society Research Scholar Award (RSG-18-009-01-CCG) to W.W., and NIH grant (R35GM130367) to R.L. Research reported in this publication was also supported in part by the UCI Interdisciplinary Pilot Study Funding in Cancer Systems Biology (U54CA217378) and the UCI Chao Family Comprehensive Cancer Center (P30CA062203) using Anti-Cancer Challenge funds.

## Author contributions

H.H. and W.W. conceived the study. R.Q. and R.L. designed the MD analysis. H.H. performed all the experiments with the assistance from G.S., J.A., B.Y., Y.L., T.L., J.Y., Y.X., D.X., J.K.Y., C.A.; Z.H., C.X and S.R. performed the MD analyses. D.A.F. helped with the CD study. Z.M. and K.-L.G. provided key reagents and commented on the manuscript; H.H., Z.H., C.X., R.L. and W.W. wrote the manuscript.

## Competing interests

K.-L.G. is a co-founder of and holds an equity interest in Vivace Therapeutics. The other authors declare no competing interests.

## Additional information

**Supplementary information** The online version contains Supplementary Material available at https://doi.org/10.1038/s41467-024-54480-y.

