## [Transparent Peer Review file · Nature Communications]

Functional annotation of the Hippo pathway somatic mutations in human cancers

Corresponding Author: Dr Wenqi Wang

Version 0:

Reviewer comments:

Reviewer #2

(Remarks to the Author)

In this comprehensive analyses of Hippo pathway missense mutations, the authors identify over 1000 mutations spanning the main Hippo pathway from TCGA and painstakingly analyses these to ask: 1. which of these are true driver mutations, 2. these extent of hippo alterations across cancers and 3. what the the likely mechanism of each of these driver mutations. The comprise as combination of genomic analysis, biochemistry experiments, cell biology analyses and protein structure prediction and testing experiments. The result is a detailed analyses of missense mutations that have been previously implicated in these pathways that have been defined as drivers. The authors should be congratulated for this detailed analyses, and yet I was surprised by the small proportion of missense mutations that were deemed to be 'drivers' based on their screens. I feel that this manuscript is well planned, executed and written (barring some minor grammatical errors that can be corrected by perhaps seeking help from an editorial review). My comments below are relatively minor and should be easily addressed in a revision:

1. Can the authors speculate on why so few of the missense mutations are drivers, and postulate if this should extend to other 'famous genes and pathways'
2. I would like to see an 'allele frequency' analysis to assess if there is a difference between drivers and passengers as to whether they are clonal or subclonal events.
3. similarly, I would like to see the association between missense mutations and structural/copy number variations in this pathways, and ask if there is a better way to identify drivers compared to passengers based on co association.
4. For the analysis for each of the gene families involved (LATS, MOB, MST etc), I would also like to see the distribution of driver versus passenger mutations across the gene and whether the site of mutations provide support as to whether mutations can be deemed driver vs passenger.
- 5 For graph 2B, its a bit awkward to put 2 different scales at either end...perhaps it should be a side by side graph or on the same scale
6. Figure 2F- a version of this pre- and post- identification of drivers would be an interesting figure to state the point here, where majority of the mutations identified were actually passengers.
- 7.MOB1A/1B xenograft assays- I would like to see the tumor growth curves on the main figure
8. can the authors speculate on why the tissue specificity for MOB1 in HNC and NF2 in schwannomas

Reviewer #3

(Remarks to the Author)

This is a rather ambitious manuscript which aims to systematically assess the involvement of the HIPPO pathway in tumorigenesis. While the proposed work is well conceived and performed it does not fully address to the hypothesis raised initially by the authors. Indeed, the manuscript provides an interesting tool to assess the molecular features of specific HIPPO mutations but it does not provide any clinical implications which therapeutic potential. This is a major issue which flaws the overall scientific value of the manuscript.

In details:

- Figure 1F is rather correlative and does not provide any clinical implications;
- In fig.1 G the authors show Hippo pathway mutations across different types of cancer. It would be more informative to analyze the association of genomic alteration rate with tumor-stage and grade in a tumor-type dependent manner as well as the possible changes in overall survival in patients bearing Hippo pathway mutations vs patients without these mutations.
- In figure 1F the authors show that mutations on MST1 and MST2, that are onco-suppressors, are mainly amplifications

instead of deletions. This is counterintuitive. Are those specific mutations relevant for tumor development in the analyzed patients? Would it be possible that those patients develop tumor because of some other mutations or pathway alterations? Can the authors analyze whether these mutations are accompanied by co-mutations or co-alterations in additional pathways?

- In figure 1H the authors show that several mutations affecting Hippo pathway components, including YAP, TAZ and TEAD genes, are missense mutations. The authors should show whether those mutations are gain of function mutations, otherwise this figure is not so informative.
- In figure 2, the authors "generate ~1,000 Hippo missense mutants for 9 major Hippo pathway components (i.e., LATS1, LATS2, MOB1A, MOB1B, MST1, MST2, MAP4K2, MAP4K3, NF2) and expressed them individually in their corresponding KO cells. Why don't they generate YAP, TAZ and TEAD mutants? If they don't generate them, they expect that mutations on YAP, TAZ and TEAD genes are not physiologically relevant.
- Are the "85 driver missense mutations for the Hippo pathway genes (Figure 2B; Table S3), which were carried by 95 cancer patients (Figure 2C; Table S3) and distributed in different types of human cancer" statistically relevant (95/1000 patients)?
- In figure 2B e 2C the usage of different scales for driver mutations vs passenger mutations is confusing since it masks the fact that passenger mutations vs driver mutations are many more (10x). Moreover, what is the biological meaning of passenger mutations if these latter don't rescue YAP cytoplasmic translocation in the relative knock-down condition? And is it correct to merge driver and passenger mutations in the statistical analyses and correlations with tumor grade, stage etc. if in the cells their expression has no effect on YAP cytoplasmic translocation?
- From Figure 2E it is again clear that the frequency of single driver mutations in specific tumors is very low within the numerous analyzed cohort (1000 PATIENTS).
- Figure 2F is not clear. Why do the authors merge Hippo and YTT mutations in the third panel? Hippo and YTT proteins have contrary functions in tumorigenesis ranging from onco-suppressive to oncogenic respectively. Moreover, the authors cannot compare the frequency of alterations at multiple genes (Hippo-related) with the frequency of alterations at single genes (oncogenic drivers such as p53, KRAS, etc).
- The authors should analyze whether patients bearing the reported mutations on NF2 gene exhibit an hyperactivation of VANGL-JNK pathway

Reviewer #4

(Remarks to the Author)

The title of this MS well delivers the significance of this work, that is a systematic attempt to provide a functional annotation of hippo pathway mutations in human cancer. This is a useful resource.

The bioassay is straightforward: they tested 1000 missense mutations for 9 members of the Hippo cascade, and asked which of these individual mutant cDNA was able to rescue YAP cytoplasmic localization in KO 293 cells. For example, MST mutants in MST KO cells, and so forth. I confess that this is clear from the figures but not simply and clearly explained in the text. Also unclear is the selection of the mutants. This should be mainstream and not hidden in a supplementary table. They run too quickly in the experimental setting and this is detrimental for the non-specialistic audience. I think this part of the text should be substantially restructured.

Fig 1 and 2 are too dense. Fig. 3-4 may (at least 1) go as supplementary figures.

They define as "driver" any missense cDNA that cannot rescue YAP cytoplasmic localization in the corresponding KO cell line; and passengers those that do rescue it.

First: There is a potential flaw in this strategy, as real hits are the products of a negative result. The latter can be the product of a real requirement, or merely due to technicalities, such as poor expression level, defective stability, unfolding, or artifacts associated etc. The risk of an exaggerated estimate of functionally relevant mutants is clear and should be taken into account, at least demonstrating for all constructs (or a substantial fraction of them $\geq 50\%$) that, upon stable transfection, each mutant protein is expressed at comparable level to non-mutant or to passenger mutants.

Second, and related to the bioassay of figure 2, it is unclear what are the criteria adopted to score as passenger or driver. They show just one little pair of pictures in which one transfected cell is controlled by non-transfected cells in the neighborhood. We definitely must see much larger set of figures for each individual set of hippo family mutants. For example cells may score positive or negative depending on scoring parameters (YAP evenness between cytoplasm and nucleus?) and/or the local state of cell density. There are very subtleties that may undermine the interpretation of the results.

Third, they should change the wording passenger and driver. These words have obvious historical implications as drivers of cancer initiation/progression. With clear functional implications for cell transformation and tumorigenesis. Here they adopt these words for non-rescuing vs. rescuing activity in their specific bioassay. All in all, this is quite misleading and should be corrected.

In the same vein, the comparison with classically established Ras/PI3K mutations having similar frequency of Hippo "driver" mutations should be revised because potentially flawed by the first point above and misled by the "driver" word, mixing layers that have little to do with each other.

Connected to the slipperiness of some of these claims is figure 6E on the MOB-defective Cal27 cell line. The entire data is driven by the transfection with control vector. That for unknown reason is boosting growth. Without this confounding variable, however, adding back MOB to Cal cells has no substantial effects, in fact indicating that MOB is not a driver mutation in these cells. So, when a MOB mutant is behaving differently than Vector (inhibiting the effects of the control plasmid), I have

really no idea on how to interpret these results. They should find an independent model system (different cellular model) to prove in vivo that restoration of Hippo proteins, (ideally) LATS or MST as paradigm hippo kinases, are effective at halting in vivo growth in corresponding mutant cell lines in a manner independent from the idiosyncrasy of a control plasmid. And demonstrate that their "driver"/relevant/functionally inactivating mutation is not effective in the same assay.

Also missing is the cross-control between different hippo pathway mutants (not for all of course, but for the few most intriguing one here identified and better validated in the revised MS). For example, MST and LATS may have YAP independent function (MST and apoptosis for example) and non-Hippo regulations. To be defined as hippo pathway mutations or not (and interesting in both cases), one would like to verify that a passenger mutation fails to rescue YAP in 293 KO for a downstream element of the hippo cascade. MST and MAP4K passenger mutants in LATS KO cells for example. or LATS passenger mutants in YAP-TEAD fusion expressing cells.

Version 1:

Reviewer comments:

Reviewer #2

(Remarks to the Author)

Thank you for your response, clarifications and modifications. I am satisfied with the revised manuscript.

Reviewer #3

(Remarks to the Author)

The authors have adequately addressed to the concerns raised previously by this reviewer. Based on this, the manuscript warrants acceptance for publication in Nat Comm.

Reviewer #4

(Remarks to the Author)

The MS has improved. however, some requests that found addressed significant weaknesses have not been addressed. My prior request: To be defined as hippo pathway mutations or not (and interesting in both cases), one would like to verify that a passenger mutation fails to rescue YAP in 293 KO for a downstream element of the hippo cascade. MST and MAP4K passenger mutants in LATS KO cells for example. or LATS passenger mutants in YAP-TEAD fusion expressing cells. I was asking about rescuing or not YAP activity and not LC3, YAP independent functions.

This key point has not been addressed.

Also unaddressed is the request of using an independent cellular model system. My original request was They should find an independent model system (different cellular model) to prove in vivo that restoration of Hippo proteins, (ideally) LATS or MST as paradigm hippo kinases, are effective at halting in vivo growth in corresponding mutant cell lines in a manner independent from the idiosyncrasy of a control plasmid. And demonstrate that their "driver"/relevant/functionally inactivating mutation is not effective in the same assay.

This has not been carried out.

Finally: related to the bioassay of figure 2. I still would like to see much more pictures to sample the real situation and quantification criteria.

Version 2:

Reviewer comments:

Reviewer #4

(Remarks to the Author)

The authors have carried out minimal revision but I understand their difficulties. All in all this is a sound paper for NCOMM

REVIEWER COMMENTS

Reviewer #2 (Remarks to the Author): *expertise in head and neck cancer in vivo models*

In this comprehensive analyses of Hippo pathway missense mutations, the authors identify over 1000 mutations spanning the main Hippo pathway from TCGA and painstakingly analyses these to ask: 1. which of these are true driver mutations, 2. these extent of hippo alterations across cancers and 3. what the the likely mechanism of each of these driver mutations. The comprise as combination of genomic analysis, biochemistry experiments, cell biology analyses and protein structure prediction and testing experiments. The result is a detailed analyses of missense mutations that have been previously implicated in these pathways that have been defined as drivers. The authors should be congratulated for this detailed analyses, and yet I was surprised by the small proportion of missense mutations that were deemed to be 'drivers' based on their screens. I feel that this manuscript is well planned, executed and written (barring some minor grammatical errors that can be corrected by perhaps seeking help from an editorial review). My comments below are relatively minor and should be easily addressed in a revision:

Thanks! We appreciate the reviewer's positive comments on this study.

During the revision, we have changed the terms “driver” and “passenger” mutations to “loss-of-function (LOF)” and “neural” mutations, respectively, to more accurately reflect our findings. We will use these updated terms to address the critiques as raised below.

1. Can the authors speculate on why so few of the missense mutations are drivers, and postulate if this should extend to other 'famous genes and pathways'

Thanks for the question! We are also surprised by the limited number of loss-of-function (LOF) mutations identified in the major Hippo pathway genes.

First, this issue may be due to the stringent strategy used in our Hippo LOF mutation screen. We re-expressed each Hippo gene missense mutant in their corresponding knockout (KO) cells and used the clear YAP nuclear-to-cytoplasmic translocation as a readout to determine their functions in the Hippo pathway. This approach may miss the mutations that partially affect the Hippo pathway genes' activities and those affecting protein stability.

Second, the Hippo pathway can be affected by different alterations in cancers (e.g., gene fusion, frameshift, splicing) (**Fig. 1f, h**) and is not limited to missense mutation. Moreover, dysregulation of Hippo pathway regulators, such as *GNAQ/GNA11* in uveal melanoma and *PTPN14* in basal cell carcinoma, also leads to Hippo pathway inactivation in cancer. These facts highlight the complex reasons for Hippo pathway inactivation in human cancer development, making it different from other cancer-related genes like *TP53*, *PI3KCA*, *KRAS*, where missense mutation are the predominant driver alterations.

We have included more discussions about these points in the revised manuscript.

2. *I would like to see an 'allele frequency' analysis to assess if there is a difference between drivers and passengers as to whether they are clonal or subclonal events.*

Since the Hippo gene LOF mutation rate is low in cancers (**Fig. 2f**), we did not conduct additional allele frequency analysis for the identified Hippo LOF and neural mutations.

3. *similarly, I would like to see the association between missense mutations and structural/copy number variations in this pathways, and ask if there is a better way to identify drivers compared to passengers based on co association.*

Thanks for the suggestion! We have analyzed the association between the number of Hippo gene mutations (i.e., missense mutation, other mutation, no mutation) and their copy variations (i.e., deletion, amplification, no change). As shown in the “**Copy number analysis**” sheet of the revised **Supplementary Table 3**, this association is statistically significant only for some Hippo pathway genes (i.e., MST2, SAV1, MAP4K3, MAP4K4, MAP4K5, MAP4K7, MOB1A, MOB1B, YAP, TAZ, TEAD1, and TEAD4). Therefore, this analysis may not fully facilitate the identification of Hippo LOF mutations.

4. *For the analysis for each of the gene families involved (LATS, MOB, MST etc), I would also like to see the distribution of driver versus passenger mutations across the gene and whether the site of mutations provide support as to whether mutations can be deemed driver vs passenger.*

As suggested by the reviewer, we have included the LOF mutation distribution information for each Hippo pathway gene in the revised **Supplementary Table 3**, where LOF mutations are highlighted among all screened missense mutations. As shown in the “**LOF mutation residue analysis**” sheet in the revised **Supplementary Table 3**, MST2-D146, MAP4K3-D196, LATS1-R694/E753/G787/R995, and LATS2-G675/A881/T1041 were identified as essential residues for these Hippo genes, because mutating them to different amino acids consistently inhibited their respective gene functions in the Hippo pathway. In contrast, MST1-D97, MST2-S106, MAP4K3-R279, LATS1-P1028 and LATS2-L967 required specific amino acid mutations to inhibit their corresponding gene functions in the Hippo pathway. This analysis shows the critical roles of specific residues in regulating the activities of Hippo pathway genes, supporting the LOF roles for these residues-associated mutations.

5 *For graph 2B, its a bit awkward to put 2 different scales at either end...perhaps it should be a side by side graph or on the same scale*

Agree! Please see the revised **Fig. 2b-e**.

6. *Figure 2F- a version of this pre- and post- identification of drivers would be an interesting figure to state the point here, where majority of the mutations identified were actually passengers.*

Thanks for the suggestion! We have included the pre-identified oncogenic alterations for the Hippo pathway genes based on TCGA and compared them to those identified in our screen study

(**Supplementary Fig. 3d** and **Supplementary Table 4**). Our study reveals the LOF missense mutations in the Hippo pathway genes that are not available in TCGA.

7. MOB1A/IB xenograft assays- I would like to see the tumor growth curves on the main figure

We attempted to measure the tongue tumor size during the orthotopic xenograft experiment. Unfortunately, we observed increased mouse death when the mice were anesthetized every other day for tumor size measurement. Therefore, we only measured the tumor size at the endpoint.

8. can the authors speculate on why the tissue specificity for MOB1 in HNC and NF2 in schwannomas

Our study identified only one head and neck cancer (HNC) patient sample carrying the MOB1 LOF mutation, making it hard to draw tissue specific conclusion regarding MOB1 mutation. In contrast, neurofibromatosis type 2-associated cancers, such as schwannomas, meningiomas, and ependymomas, consistently exhibit NF2 deficiency due to its gene deletions and mutations. Our findings here align with previous studies, highlighting the tumor suppressor role of NF2 in these types of cancers.

Reviewer #3 (Remarks to the Author): expertise in Hippo genomics

This is a rather ambitious manuscript which aims to systematically assess the involvement of the HIPPO pathway in tumorigenesis. While the proposed work is well conceived and performed it does not fully address to the hypothesis raised initially by the authors. Indeed, the manuscript provides an interesting tool to assess the molecular features of specific HIPPO mutations but it does not provide any clinical implications which therapeutic potential. This is a major issue which flaws the overall scientific value of the manuscript.

We thank the reviewer for the positive comments on our study!

In terms of clinical implication, we analyzed Hippo signaling alterations using TCGA data. Our findings demonstrated that cancer patient samples with altered Hippo genes exhibit high histologic grade (**Fig. 1c** and **Supplementary Table 1**), advanced tumor stage (**Fig. 1d** and **Supplementary Table 1**), and increased disease stage (**Fig. 1e** and **Supplementary Table 1**). In addition, cancer patients with altered Hippo genes showed poor overall survival rate (**Supplementary Fig. 2d**). As stated in the manuscript, the major goals of our study are to **1**) provide a comprehensive overview of Hippo pathway dysregulations in human cancers and **2**) functionally characterize cancer patients-associated missense mutations in regulating the Hippo pathway. We believe that our study will significantly advance our understanding of Hippo pathway dysregulations in cancer development.

During the revision, we have changed the terms “driver” and “passenger” mutations to “loss-of-function (LOF)” and “neural” mutations, respectively, to more accurately reflect our findings. We will use these updated terms to address the critiques as raised below.

In details:

- *Figure 1F is rather correlative and does not provide any clinical implications;*

Thanks for pointing out this issue! We analyzed the clinical association for individual Hippo signaling genes in TCGA. As shown in **Supplementary Fig. 1a** and **Supplementary Table 1**, cancer patient samples with alterations in individual Hippo signaling genes mostly exhibited higher histologic grade. However, only a few Hippo genes showed significant correlations with advanced tumor stage (**Supplementary Fig. 1b** and **Supplementary Table 1**) and increased disease stage (**Supplementary Fig. 1c** and **Supplementary Table 1**). These relatively low clinical correlation for individual Hippo signaling genes may be due to their low somatic alteration rate in human cancers (**Fig. 1f**). These data highlight the complex and variable roles of Hippo signaling genes in cancer and suggest that further research is needed to fully understand their contributions to cancer development.

- *In fig.1 G the authors show Hippo pathway mutations across different types of cancer. It would be more informative to analyze the association of genomic alteration rate with tumor-stage and grade in a tumor-type dependent manner as well as the possible changes in overall survival in patients bearing Hippo pathway mutations vs patients without these mutations.*

Thanks for the suggestions! We analyzed the Hippo signaling alterations in different types of cancers in TCGA and found that only several cancer types have tumor stage and grade information available. As shown in **Supplementary Fig. 2a** and **Supplementary Table 1**, bladder urothelial carcinoma (BLCA) patient samples with Hippo signaling alterations exhibited high histological grades. Prostate adenocarcinoma (PRAD) patient samples with altered Hippo signaling genes were significantly correlated with advanced tumor stages (**Supplementary Fig. 2b** and **Supplementary Table 1**). Kidney chromophobe (KICH) and kidney renal papillary cell carcinoma (KIRP) are two types of cancers showing significant correlations between altered Hippo signaling genes and increased disease stages (**Supplementary Fig. 2c** and **Supplementary Table 1**). Interestingly, colon adenocarcinoma (COAD) cancer patient samples with Hippo signaling alterations exhibited decreased disease stages (**Supplementary Fig. 2c** and **Supplementary Table 1**).

Moreover, we found that cancer patients with Hippo signaling alterations showed poor overall survival rate as compared to those with normal Hippo signaling genes (**Supplementary Fig. 2d**). Specifically, esophageal carcinoma (ESCA), kidney chromophobe (KICH), kidney renal clear cell carcinoma (KIRC), liver hepatocellular carcinoma (LIHC), and prostate adenocarcinoma (PRAD) patients with altered Hippo signaling genes exhibited poor overall survival (**Supplementary Fig. 2e**). In contrast, Hippo signaling gene alterations indicate better overall survival for bladder urothelial carcinoma (BLCA) and glioblastoma (GBM) patients (**Supplementary Fig. 2e**).

Collectively, these findings suggest the complex and context-dependent nature of Hippo signaling alterations in different cancers, underscoring the need for personalized therapeutic approaches based on specific cancer types and genetic contexts.

- *In figure 1F the authors show that mutations on MST1 and MST2, that are onco-suppressors,*

are mainly amplifications instead of deletions. This is counterintuitive. Are those specific mutations relevant for tumor development in the analyzed patients? Would it be possible that those patients develop tumor because of some other mutations or pathway alterations? Can the authors analyze whether these mutations are accompanied by co-mutations or co-alterations in additional pathways?

Thanks for pointing out this issue! MST2 and MAP4K7 are two Hippo pathway components that are amplified in human cancers (**Fig. 1f**). Although they are generally considered as tumor suppressors due to their roles in the Hippo pathway, MST2 and MAP4K7 also have YAP/TAZ-independent functions that drive cancer development. For example, MST2 is amplified in gastric cancer, where it induces gastric carcinogenesis by activating the Ras-MAPK pathway (PMID: 34772410). MST2 is also amplified in prostate cancer and is required for prostate cancer cell proliferation and invasion (PMID: 34450249). MAP4K7 is highly expressed in lung squamous cell carcinoma (LSCC), promoting LSCC cell proliferation and survival through FAK (PMID: 33495197). These facts are consistent with our findings (**Fig. 1f**), suggesting the complex roles of altered Hippo pathway genes in human cancer development. We have included more discussions on the amplification of MST2 and MAP4K7 in the revised manuscript.

• In figure 1H the authors show that several mutations affecting Hippo pathway components, including YAP, TAZ and TEAD genes, are missense mutations. The authors should show whether those mutations are gain of function mutations, otherwise this figure is not so informative.

Thanks for pointing out this issue! A previous study has already investigated the reported missense mutations for YAP and TAZ (PMID: 30380420), so we did not repeat this analysis. Instead, we included the findings from this study into our oncogenic alteration analysis (**Supplementary Fig. 3d, e and Supplementary Table 4**).

• In figure 2, the authors “generate~1,000 Hippo missense mutants for 9 major Hippo pathway components (i.e., LATS1, LATS2, MOB1A, MOB1B, MST1, MST2, MAP4K2, MAP4K3, NF2) and expressed them individually in their corresponding KO cells. Why don’t they generate YAP, TAZ and TEAD mutants? If they don’t generate them, they expect that mutations on YAP, TAZ and TEAD genes are not physiologically relevant.

We thank the reviewer for raising this important question. Our current study majorly focuses on the Hippo pathway genes to determine how they become dysregulated and how their dysregulation leads to Hippo pathway inactivation. Notably, a previous study has already characterized the cancer-associated somatic mutations of YAP and TAZ (PMID: 30380420). Therefore, we did not perform a similar gain-of-function study but instead incorporated the findings from this published work into our current oncogenic alteration analysis (**Supplementary Fig. 3d, e and Supplementary Table 4**).

• Are the “85 driver missense mutations for the Hippo pathway genes (Figure 2B; Table S3), which were carried by 95 cancer patients (Figure 2C; Table S3) and distributed in different types of human cancer” statistically relevant (95/11000 patients)?

Thanks for the question! To answer it, we analyzed the number of patients with the identified Hippo LOF mutations in various types of cancer (**Fig. 2f**). As shown in the revised “**Summary**” sheet of **Supplementary Table 3**, Hippo LOF missense mutations are significantly associated with bladder urothelial carcinoma (BLCA), mesothelioma (MESO), skin cutaneous melanoma (SKCM) and uterine corpus endometrial carcinoma (UCEC). These findings suggest that Hippo LOF missense mutations may play a critical role in the development and/or progression of these cancer types.

• In figure 2B e 2C the usage of different scales for driver mutations vs passenger mutations is confusing since it masks the fact that passenger mutations vs driver mutations are many more (10x). Moreover, what is the biological meaning of passenger mutations if these latter don't rescue YAP cytoplasmic translocation in the relative knock-down condition? And is it correct to merge driver and passenger mutations in the statistical analyses and correlations with tumor grade, stage etc. if in the cells their expression has no effect on YAP cytoplasmic translocation?

Thanks for pointing out these issues! We have revised these figures to better present our Hippo missense mutation screen results. Please see the revised **Fig. 2b-e**.

In this study, we defined the Hippo neutral mutations based on the criterion that they did not affect the ability of Hippo pathway genes to rescue YAP cytoplasmic localization in their corresponding KO cells. However, this approach failed to reveal potential neutral mutations that could activate the Hippo pathway in specific cancer types where it functions as an oncogenic pathway, such as hematological cancers, ER+ breast cancer, small cell lung cancer. Additionally, neutral mutations may impact the roles of Hippo pathway components in YAP-independent, cancer-relevant events. For example, our recent study reveals that MAP4K2 binds and phosphorylates LC3 to regulate autophagy in energy stress conditions (PMID: 37595580). Interestingly, one identified neutral mutation, W356R, is located in the LC3-interacting region (LIR) motif (**Supplementary Fig. 4d**) of MAP4K2 and can disrupt its interaction (**Supplementary Fig. 4e**) and co-localization with LC3 (**Supplementary Fig. 4f**), suggesting the inhibitory effect of this neutral mutation on MAP4K2-mediated autophagy. Based on these facts, we think that it would be hard to conclude the non-tumorigenic roles of the identified Hippo neutral mutations, thus chose to use all the TCGA-documented alterations of the Hippo signaling genes for our clinical analyses (**Fig. 1c-e** and **Supplementary Fig. 1, 2**).

• From Figure 2E it is again clear that the frequency of single driver mutations in specific tumors is very low within the numerous analyzed cohort (11000 PATIENTS).

Agree! We believe this issue could be caused by the stringent criterion used in our Hippo loss-of-function (LOF) mutation screen. To achieve this, we re-expressed each Hippo gene missense mutant in the corresponding KO cells and used the clear YAP nuclear-to-cytoplasmic translocation as a readout to determine their functions in the Hippo pathway. This strategy may result in missing weak LOF mutations that partially affect the Hippo pathway activity and potential LOF mutations that affect protein stability due to the overexpression approach used here. We have included more discussions about these potential issues in the revised manuscript.

• *Figure 2F is not clear. Why do the authors merge Hippo and YTT mutations in the third panel? Hippo and YTT proteins have contrary functions in tumorigenesis ranging from onco-suppressive to oncogenic respectively. Moreover, the authors cannot compare the frequency of alterations at multiple genes (Hippo-related) with the frequency of alterations at single genes (oncogenic drivers such as p53, KRAS, etc).*

The oncogenic alterations in the Hippo pathway lead to its inactivation, resulting in the activation of its downstream YAP/TAZ/TEAD (YTT). Conversely, the oncogenic alterations of YAP/TAZ/TEAD can independently activate their transcriptional activities. Therefore, both Hippo and YTT oncogenic alterations are functionally consistent in driving cancer development. Since cancer patients with Hippo oncogenic alterations differ from those with YAP/TAZ/TEAD oncogenic alterations, we combined the Hippo pathway and YTT to represent the entire cohort of cancer patients with oncogenic alterations in Hippo signaling genes.

We acknowledge the potential concern of grouping multiple Hippo pathway genes together for comparison with individual cancer-related genes (i.e., TP53, PI3KCA, PTEN, KRAS, HRAS). Here, we used this approach given the consistent roles of the Hippo pathway genes in inhibiting YAP/TAZ during cancer development. We appreciate the reviewer critique on this point and have moved these figures to the supplemental materials as **Supplementary Fig. 3d, e**. Moreover, we have revised “Driver alteration” to “Oncogenic alteration” in these figures, as there is no direct evidence supporting the identified LOF missense mutations as drivers for cancer initiation and progression.

• *The authors should analyze whether patients bearing the reported mutations on NF2 gene exhibit an hyperactivation of VANGL-JNK pathway*

Thanks for the suggestion! Elevated JNK activity has been observed in NF2-deficient primary human schwannoma cells (PMID: 12761036), although the underlying mechanism remains unclear. Our current study has provided potential mechanistic insights into this correlation, which will be further investigated in our future research.

Reviewer #4 (Remarks to the Author): expertise in Hippo signalling

The title of this MS well delivers the significance of this work, that is a systematic attempt to provide a functional annotation of hippo pathway mutations in human cancer. This is a useful resource. The bioassay is straightforward: they tested 1000 missense mutations for 9 members of the Hippo cascade, and asked which of these individual mutant cDNA was able to rescue YAP cytoplasmic localization in KO 293 cells. For example, MST mutants in MST KO cells, and so forth.

Thanks for the nice summary and positive comments on our work!

During the revision, we have changed the terms “driver” and “passenger” mutations to “loss-of-function (LOF)” and “neural” mutations, respectively, to more accurately reflect our findings. We will use these updated terms to address the critiques as raised below.

I confess that this is clear from the figures but not simply and clearly explained in the text. Also unclear is the selection of the mutants. This should be mainstream and not hidden in a supplementary table. They run too quickly in the experimental setting and this is detrimental for the non-specialistic audience. I think this part of the text should be substantially restructured.

Sorry for the confusions! All the TCGA-documented Hippo gene missense mutations were generated and used for the screening study. We have included more details about the related experimental settings in the revised manuscript.

Fig 1 and 2 are too dense. Fig.3 4 may (at least 1) go as supplementary figures.

Thanks for the suggestions! We have moved some figures of **Fig. 2** to the supplemental materials (**Supplementary Fig. 3d, e**). In addition, we validated and characterized the identified loss-of-function (LOF) mutations in the Hippo pathway genes LATS1/2 (**Fig. 3**), MST1/2 (**Fig. 4**), MAP4K2/3 (**Fig. 5**), MOB1A/B (**Fig. 6**), and NF2 (**Fig. 7**) in parallel. We appreciate the reviewer's thoughtful suggestion but hope to keep **Fig. 3, 4** as main figures to make a better balance between these Hippo pathway genes in this resource study.

They define as "driver" any missense cDNA that cannot rescue YAP cytoplasmic localization in the corresponding KO cell line; and passengers those that do rescue it.

First: There is a potential flaw in this strategy, as real hits are the products of a negative result. The latter can be the product of a real requirement, or merely due to technicalities, such as poor expression level, defective stability, unfolding, or artifacts associated etc. The risk of an exaggerated estimate of functionally relevant mutants is clear and should be taken into account, at least demonstrating for all constructs (or a substantial fraction of them $\geq 50\%$) that, upon stable transfection, each mutant protein is expressed at comparable level to non mutant or to passenger mutants.

Thanks for pointing out this issue! Our screen was performed using an overexpression approach, where each Hippo gene missense mutant was transiently expressed in the corresponding KO cells to examine YAP nuclear-to-cytoplasmic translocation in the mutant-expressing KO cells. Given the variety of plasmid expression conditions, we selected the cells with high mutant plasmid expression (i.e., high immunofluorescent signal of the mutant construct tag) to examine the change in cellular localization of YAP. This approach allowed us to avoid technical issues related to poor mutant expression level and defective stability. This concern is further addressed by our biochemical validation studies, where all the identified LOF mutants can be similarly detected and compared for their defects in activating the Hippo pathway (**Fig. 3-7**).

Moreover, we generated cells stably expressing LATS1, LATS2, MST2, MOB1A/B, and NF2 LOF mutants and compared their expression to wild-type proteins. As shown in **Supplementary Fig. 4a-c**, the LATS1, LATS2, and MST2 LOF mutant proteins were expressed at levels comparable to the wild-type proteins in stable cells. Similarly, the NF2 LOF mutations did not affect NF2 protein expression in different cell lines (**Fig. 7c, f**). In contrast, the H161Y and H161D LOF mutations significantly reduced MOB1A and MOB1B expression, respectively (data not shown), which may be due to their structural effects on MOB1A/B (**Fig. 6k-m**).

Despite this, when the expression levels of the MOB1A-H161Y and MOB1B-H161D LOF mutants were made comparable to that of wild-type MOB1A and MOB1B through multiple rounds of lentiviral infection (**Fig. 6b, d**), these MOB1 LOF mutants still failed to rescue YAP phosphorylation, reduce YAP downstream gene transcription, and inhibit xenograft tumor growth of the MOB1A/B double KO cells (**Fig. 6b-f**).

Second, and related to the bioassay of figure 2, it is unclear what are the criteria adopted to score as passenger or driver. They show just one little pair of pictures in which one transfected cell is controlled by non-transfected cells in the neighborhood. We definitively must see much larger set of figures for each individual set of hippo family mutants. For example cells may score positive or negative depending on scoring parameters (YAP evenness between cytoplasm and nucleus?) and/or the local state of cell density. There are very subtleties that may undermine the interpretation of the results.

Apologies for missing additional details about our mutation screen! In our screen, Hippo KO cells were transfected with each Hippo gene mutant construct overnight, serum starved for 5-6 hours (i.e., to increase the Hippo pathway activity), and subjected to immunofluorescence. Wild-type Hippo gene and its known inactive mutant (e.g., kinase dead mutant) were included as positive and negative controls, respectively.

Regarding the criteria, if the mutant protein induced a clear YAP nuclear-to-cytoplasmic translocation similar to wild-type Hippo protein, we concluded that this mutation was a neutral mutation. If the mutant protein did not affect nuclear localization of YAP, like the known Hippo inactive mutant protein, we classified the mutation as a LOF mutation. We agree that this stringent criterion may result in missing weak LOF mutations that partially affect Hippo pathway activity and potential LOF mutations that affect protein stability due to the overexpression approach used here. We have included more discussions about these issues in the revised manuscript.

Third, they should change the wording passenger and driver. These words have obvious historical implications as drivers of cancer initiation/progression. With clear functional implications for cell transformation and tumorigenesis. Here they adopt these words for non-rescuing vs. rescuing activity in their specific bioassay. All in all, this is quite misleading and should be corrected.

In the same vein, the comparison with classically established Ras/PI3K mutations having similar frequency of Hippo "driver" mutations should be revised because potentially flawed by the first point above and misled by the "driver" word, mixing layers that have little to do with each other.

Agree! We have revised the terms “driver” and “passenger” mutations to “loss-of-function (LOF)” and “neutral” mutations, respectively, as there is no direct evidence supporting the identified LOF missense mutations as drivers of cancer initiation and/or progression. To address this point, we have moved the related figures to the supplemental materials as **Supplementary Fig. 3d, e**, and revised “Driver alteration” to “Oncogenic alteration” in these figures.

Connected to the slipperiness of some of these claims is figure 6E on the MOB-defective Cal27 cell line. The entire data is driven by the transfection with control vector. That for unknown reason is boosting growth. Without this confounding variable, however, adding back MOB to Cal cells has no substantial effects, in fact indicating that MOB is not a driver mutation in these cells. So, when a MOB mutant is behaving differently than Vector (inhibiting the effects of the control plasmid), I have really no idea on how to interpret these results. They should find an independent model system (different cellular model) to prove in vivo that restoration of Hippo proteins, (ideally) LATS or MST as paradigm hippo kinases, are effective at halting in vivo growth in corresponding mutant cell lines in a manner independent from the idiosyncrasy of a control plasmid. And demonstrate that their "driver"/relevant/functionally inactivating mutation is not effective in the same assay.

Sorry for the confusion! The increased tumor growth was actually caused by the knockout of MOB1A/B, but not by the control vector expression.

For **Fig. 6d-f**, the MOB1A/B double knockout (DKO) CAL-27 cells were stably transduced with vector control, wild-type MOB1A/B, and their LOF mutants (H161Y for MOB1A and H161D for MOB1B) and subjected to an orthotopic xenograft tumor assay. Our data suggest that vector control-transduced MOB1A/B DKO CAL-27 xenograft tumor showed increased growth compared to wild-type CAL-27 tumors (**Fig. 6d-f**). Reconstituting wild-type MOB1A/B, but not their LOF mutants, largely attenuated MOB1A/B DKO CAL-27 tumor growth (**Fig. 6d-f**). These data suggest the oncogenic roles of the MOB1 LOF mutations in promoting tumor growth.

Also missing is the cross-control between different hippo pathway mutants (not for all of course, but for the few most intriguing one here identified and better validated in the revised MS). For example, MST and LATS may have YAP independent function (MST and apoptosis for example) and non-Hippo regulations. To be defined as hippo pathway mutations or not (and interesting in both cases), one would like to verify that a passenger mutation fails to rescue YAP in 293 KO for a downstream element of the hippo cascade. MST and MAP4K passenger mutants in LATS KO cells for example. or LATS passenger mutants in YAP-TEAD fusion expressing cells.

Thanks for pointing out this issue! As discussed in the manuscript, we recognize the YAP/TAZ-independent functions of the Hippo pathway components, although the focus of our current study is primarily on classic Hippo-YAP/TAZ signaling.

We appreciate the reviewer's cross-control suggestion. To test the hypothesis, we focused on several identified neutral mutations of MAP4K2, because our recent study revealed a YAP-independent role of MAP4K2 in regulating autophagy and cell survival by binding and phosphorylating LC3 (PMID: 37595580). Interestingly, one MAP4K2 neutral mutation, W356R, was found located in its LC3-interacting region (LIR) motif (i.e., EEWTL~~L~~) (**Supplementary Fig. 4d**). In contrast to other nearby neutral mutations, W356R disrupted the interaction of MAP4K2 with LC3 in both wild-type and LATS1/2 double knockout (DKO) cells (**Supplementary Fig. 4e**). Moreover, MAP4K2 W356R mutant failed to localize onto autophagosome in chloroquine-treated wild-type and LATS1/2 DKO cells (**Supplementary Fig. 4f**). These findings suggest that the W356R neutral mutation of MAP4K2 inhibits its YAP-independent role in regulating autophagy.

Response to Reviewers

Reviewer #2 (Remarks to the Author):

Thank you for your response, clarifications and modifications. I am satisfied with the revised manuscript.

Thanks!

Reviewer #3 (Remarks to the Author):

The authors have adequately addressed to the concerns raised previously by this reviewer. Based on this, the manuscript warrants acceptance for publication in Nat Comm.

Thanks!

Reviewer #4 (Remarks to the Author):

The MS has improved. however, some requests that found addressed significant weaknesses have not been addressed.

My prior request: To be defined as hippo pathway mutations or not (and interesting in both cases), one would like to verify that a passenger mutation fails to rescue YAP in 293 KO for a downstream element of the hippo cascade. MST and MAP4K passenger mutants in LATS KO cells for example. or LATS passenger mutants in YAP-TEAD fusion expressing cells. I was asking about rescuing or not YAP activity and not LC3, YAP independent functions.

This key point has not been addressed.

Sorry for the misunderstanding! As suggested by the reviewer, we randomly selected approximately 20 neutral (passenger) mutations for the Hippo kinases MST1, MST2 and MAP4K3, and expressed them individually in the LATS1/2 double knockout (DKO) cells. In contrast to LATS1 and LATS2 (Supplementary Fig. 3b**), neither the wild-type MST/MAP4K proteins nor their neutral mutants (**Supplementary Fig. 4a-c**) were able to rescue YAP's cytoplasmic localization in the LATS1/2 DKO cells. These results show the essential role of LATS1/2 in mediating the inhibitory effect of MST/MAP4K neutral mutations on YAP.**

Also unaddressed is the request of using an independent cellular model system. My original request was

They should find an independent model system (different cellular model) to prove in vivo that restoration of Hippo proteins, (ideally) LATS or MST as paradigm hippo kinases, are effective at halting in vivo growth in corresponding mutant cell lines in a manner independent from the idiosyncrasy of a control plasmid. And demonstrate that their "driver"/relevant/functionally inactivating mutation is not effective in the same assay.

This has not been carried out.

Thanks for the suggestion! Different model systems have been utilized to confirm the oncogenic effects of MOB1 and NF2's loss-of-function (LOF) mutations. In the MOB1 and NF2-related studies, our data clearly show that the LOF mutations identified in head and neck cancer (HNC) and meningioma patients disrupted the tumor suppressor functions of MOB1 and NF2 *in vivo*. Specifically, we depleted MOB1A/B in the HNC cell line CAL-27 and NF2 in the meningioma cell line IOMM-Lee, and then reconstituted wild-type MOB1 and NF2, along with their LOF mutants, into the corresponding KO cancer cell lines. Restoring wild-type MOB1 and NF2 significantly inhibited xenograft tumor growth of the KO cells, whereas this inhibitory effect was not observed for the vector control or their LOF mutants (**Fig. 6d-f** and **7o-q**).

Finally: related to the bioassay of figure 2. I still would like to see much more pictures to sample the real situation and quantification criteria.

We screened the Hippo missense mutants by checking them individually under microscope, though we did not capture images of all of them due to the large number of mutants. Please find some representative images below.

We transiently expressed the indicated Hippo gene mutants into their corresponding KO cells. After serum starvation, cells were subjected to immunofluorescence using anti-Flag and anti-YAP antibodies. Arrows showed the cells expressing the indicated constructs.

A**B****C**
Response to Reviewers

Reviewer #4 (Remarks to the Author):

The authors have carried out minimal revision but I understand their difficulties. All in all this is a sound paper for NCOMM

Thanks!